# Lithostratigraphy, Origin, and Geodynamic Setting of Iron Formations and Host Rocks of the Anyouzok Region, Congo Craton, Southwestern Cameroon

**Isaac Swiffa Fajong** [1], **Marvine Nzepang Tankwa** [2], **Donald Hermann Fossi** [1,2], **Sylvestre Ganno** [1,*], **Cyriel Moudioh** [2], **Landry Soh Tamehe** [3], **Cheo Emmanuel Suh** [4] **and Jean Paul Nzenti** [1]

1. Department of Earth Sciences, Faculty of Science, University of Yaounde I, Yaounde P.O. Box 812, Cameroon
2. Institute for Geological and Mining Research, Yaounde P.O. Box 4110, Cameroon
3. School of Geosciences and Info-Physics, Central South University, Changsha 410083, China
4. Economic Geology Unit, Department of Geology, University of Buea, Buea P.O. Box 63, Cameroon
* Correspondence: sganno2000@gmail.com; Tel.: +237-677-752-579

**Abstract:** In Cameroon, most of the iron formation occurrences reported are found within the Nyong and Ntem Complexes. The Anyouzok iron deposit is located in the Nyong Complex greenstone belts, which represent the NW margin of this Congo craton. The main lithological units comprise the iron formations (IFs) unit, consisting of banded IFs (BIFs) and sheared BIFs (SBIFs), and the associated metavolcanic rocks unit consisting of mafic granulite, garnet amphibolite, and biotite gneiss. Within the Anyouzok area, BIFs are rare, while SBIFs are ubiquitous. This study reports the petrography, mineralogy, and whole rock geochemistry of IFs and interbedded metavolcanic rocks of the Anyouzok iron deposit. The abundance of cavities, higher Fe contents (49.60–55.20 wt%), and strong Eu anomalies (Eu/Eu* = 2.14–3.17) within the SBIFs compared to the BIFs suggest that SBIFs were upgraded through post-depositional hydrothermal alteration activities. REE signatures indicate the contribution of both seawater and hydrothermal fluids during BIFs precipitation. Mafic granulite and garnet amphibolite protoliths were derived from the partial melting of a metasomatized spinel lherzolite depleted mantle source. The overall compositional variations of the Anyouzok IFs and interbedded metavolcanic rocks endorse an Algoma-type formation deposited in the back-arc basin under suboxic to anoxic conditions.

**Keywords:** BIFs; seawater; hydrothermal fluids; back-arc/arc settings; Nyong Complex greenstone belts; Anyouzok; Congo craton

## 1. Introduction

Iron formations (IFs) are Precambrian sedimentary rocks, typically thin-bedded or laminated, containing 15% or more iron of sedimentary origin and commonly but not necessarily containing chert layers [1]. Iron ore is an essential raw material for several industrialization products, especially for developing nations. Despite the increasing demand for high-grade (>55 wt% Fe), high-purity (e.g., low phosphorus) Fe concentrates, hematite/goethite-rich ore bodies are increasingly difficult to find. Therefore, industries and policymakers are getting more and more interested in low-grade iron ore [2]. Exploration efforts are thus geared towards Archaean and Paleoproterozoic IFs-hosted magnetite-rich deposits [2,3]. Several exploration works have been carried out on the Anyouzok iron ore deposit [4,5], the latest of them being the prefeasibility studies conducted by Caminex Sarl, the Cameroonian subsidiary of the British-based International Mining and Infrastructure Corporation (IMIC). These studies reported magnetite ore deposits, with 96.9 Mt at 34.92% Fe indicated and 79.4 Mt at 35.04% Fe inferred [5]. Besides their high economic value in the steel and construction industries, IFs provide invaluable information in the understanding of the evolution of the atmosphere, biosphere, and coeval ocean composition,

as well as the origin and growth of continents [6–15]. In spite of the extensive studies in the last century, many controversies still persist concerning their origin as well as regarding how these formations are upgraded to iron ore [2,3,7,16,17]. Early classifications divided IFs into Algoma- and Superior-type [18]. Superior-type IFs are extensive, closely associated with clastic to carbonate rocks, and were deposited in near-shore continental shelf environments with no direct correlation with volcanic rocks [18]. Conversely, Algoma-type IFs are less extensive, closely affiliated with volcanic rocks in greenstone belts, and generally deposited in intracratonic rifts or in back-arc/arc basins [18]. Texturally, iron formations were also divided into two groups: banded iron formation (BIFs), widespread in Archaean to early Paleoproterozoic successions, and granular iron formations (GIFs), much more common in Paleoproterozoic successions [7].

In Cameroon, IFs are mainly found within the Nyong and Ntem Complexes (Figure 1), which correspond to the Northwestern extension of the Congo Craton [19]. The Nyong Complex (Figure 1), where the Anyouzok iron ore deposit lies, hosts several greenstone belts mainly comprising metavolcanic-sedimentary rocks associated with IFs [11,12,14,19–31]. It constitutes an emerging iron ore province of south Cameroon. This complex has experienced deformation and high-grade metamorphism, rendering the reconstruction of the depositional environment of the hosted IFs difficult. As a palliative, several workers worldwide have investigated interbedded igneous and/or sedimentary rocks to better constrain IFs' depositional setting [9–12,14,28]. In this regard, the present study provides a comprehensive geochemical dataset for IFs and interbedded metavolcanic rocks, which were intercepted in drillholes of the Anyouzok iron deposit (Figure 2a). Lithostratigraphy and petrography are presented, in combination with bulk-rock major, trace, and rare earth elements (REE) geochemistry, with the aim of determining the origin and depositional environment of the Anyouzok IFs, which is important for the understanding of the Nyong Complex geodynamic evolution.

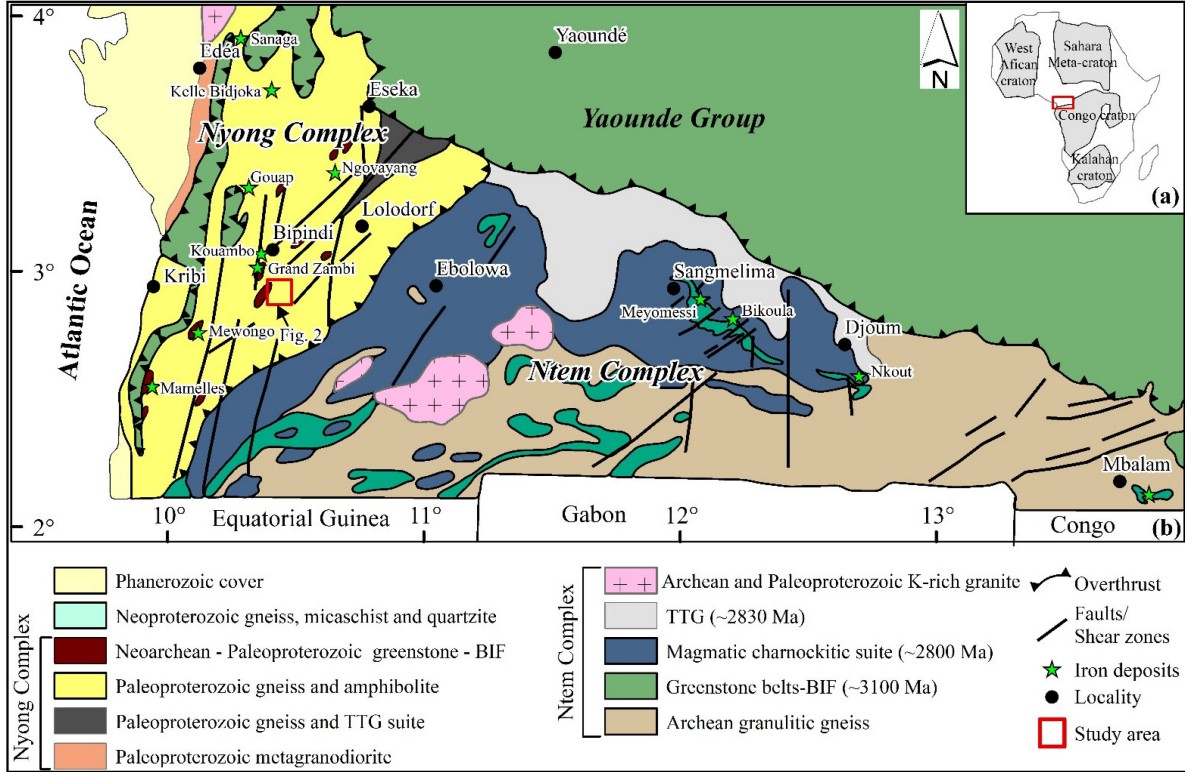

**Figure 1.** Sketch geological map of SW Cameroon (modified after [19], with insert showing the Congo craton in relation to other African cratons.

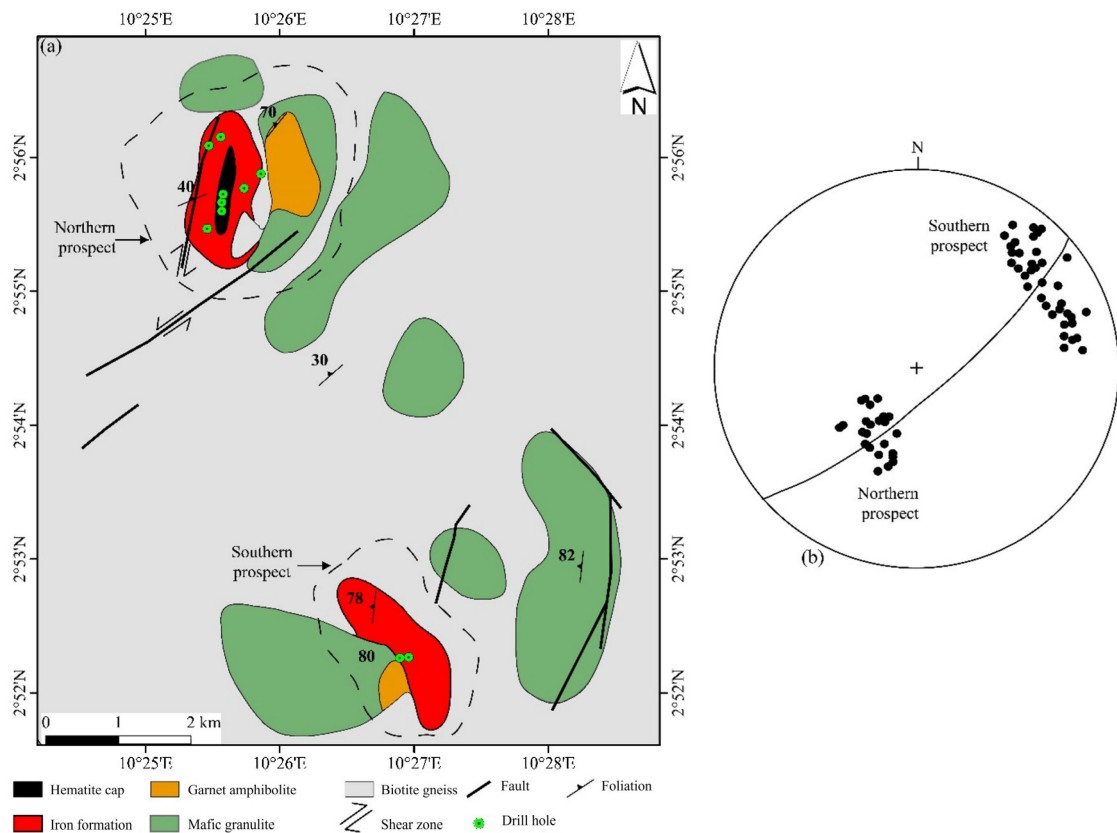

**Figure 2.** (**a**) Geological map of the Anyouzok iron deposit with drillholes and prospects (northern and southern) locations; (**b**) orientation diagram presenting the $S_1/S_2$ foliations of the Anyouzok northern and southern prospects.

## 2. Geological Setting

### 2.1. Regional Geology

Pioneer studies reported that the NW extension of the Congo Craton (CC) in Cameroon is represented by the Nyong and Ntem Complexes (Figure 1) [19]. Subsequent geophysical investigations reported that the cratonic basement in Cameroon extends northwards to the Adamawa Yade region, as revealed by the existence of high-gravity anomalies from denser material beneath this region [32–34]. These results are consistent with recent petrological investigations and U-Pb on zircon LA-ICP-MS [35], LA-MC-ICP-MS, and SHRIMP dating [36], suggesting an extension of the northern edge of the Congo Craton in the Central domain of the Pan-African North Equatorial fold belt and eastward to Central African Republic.

The Anyouzok iron ore deposit is located within the Nyong Complex (Figure 1), which is considered as a segment of the Archean Ntem Complex that was reactivated during the Paleoproterozoic Eburnean/Trans-Amazonian orogeny [14,37–39] or as a Paleoproterozoic suture zone contemporaneous to a nappe tectonic event between the São Francisco and Congo Cratons [20,23,24,40,41]. The Nyong Complex consists of various gneisses, micaschists, amphibolites, IFs, metagranodiorites, charnockites, dolerites, quartzites, tonalite-trondhjemite-granodiorite suite, syenites, serpentinites, and eclogites [11,12,14,20,22–24,28,30,31,42,43].

Few geochronological investigations characterized the geodynamic evolution of the Nyong Complex during Precambrian times [12,14,27,29,31,35,39,43,44]. LA-ICP-MS U-Pb on zircon dating of the Nyong Complex metabasic rocks (amphibolites) yielded Archean ages of 3072 ± 28 Ma [45] and 2819 ± 12 Ma [46], interpreted as the crystallization age of their precursor. The Neoarchaean age (2699 ± 7 Ma) obtained from SHRIMP zircon U-Pb isotope data on magnetite gneiss (IFs) has been interpreted as the onset age of IFs deposition in the Nyong Complex [47]. Few workers using SHRIMP U–Pb on zircon analyses [20]

and LA-ICP-MS U-Pb on zircon [27] from metasediments, constrained the maximum depositional age of the Nyong Complex at ca. 2423 Ma. Owona et al. [39], using LA-ICP-MS U-Pb on zircon suggested that the Nyong Complex metasediments have a depositional age bracketed between 2400 and 2200 Ma. Paleoproterozoic ages between 2000 and 2100 Ma have been widely reported within the Nyong Complex [12,20,27,29,43,44]. These ages have been linked to a high-grade metamorphic event and metasomatism, to the tectonic emplacement of plutonic rocks, or to an eclogite facies metamorphism. More recently, Soh Tamehe et al. [14] combining SIMS and LA-ICP-MS U-Pb analyses on zircon from the Gouap metasiliciclastic rocks constrained the depositional age of a BIF sequence at 2100–2000 Ma. These Neoarchaean and Paleoproterozoic formations generally bear Neoproterozoic imprints (600–500 Ma), corresponding to later Pan-African tectonometamorphic activities [12,43,47–49].

The Nyong Complex exhibits shallow dipping $S_1/S_2$ foliations with variably oriented axial fold (N-S to NNE-SSW) and stretching (E-W to NW-SE) lineations and local large open folds associated with N-S sinistral strike slip faults [19,20,43,50].

*2.2. Anyouzok Deposit Geology*

The Anyouzok iron ore deposit is found between longitudes 10°24′12″ E to 10°28′54″ E and latitudes 2°51′38″ N to 2°56′51″ N, covering a total surface area of about 80 km$^2$ (Figure 2a). Field investigations revealed that the study area consists of banded IFs (BIFs) associated with mafic granulite, garnet amphibolite, biotite gneiss, and mylonitized gneiss. All BIFs encountered on the field are sheared and will be called Sheared BIF (SBIF) in this study. SBIFs occur as road cuttings and on river beds (Figure 3a,b) generally associated with mafic granulite (Figure 3b). Previous workers reported that rocks within this region have undergone greenschist to granulite facies metamorphism [11,21]. From a combination of Landsat image processing, field mapping, and geostatistical analysis carried out around the neighboring Abiete-Toko gold district, this area is known to have undergone a ductile/brittle polyphase deformation, $(D_1–D_3)$, represented by the $S_1$ foliation/schistosity, $L_1$ lineation, $S_2$ foliation, and $F_2$ folds with $F_3$ shear zones and faults. The second deformation phase, $D_2$, is dominant, characterized by regular folds. It is therefore suggested that there was an N-S and NE-SW shortening direction, as expressed by the folds and localized strike-slip shear zones [51].

Based on IFs occurrences on the field, the Anyouzok iron deposit is subdivided into a northern and a southern prospect (Figure 2a). The northern prospect is an N-S trending ore body with a strike length of 2100 m by 800 m (Figure 2a). The southern prospect is an NW-SE trending ore body with a strike-length of 2000 m by 300 m (Figure 2a). In these prospects, compositional layering $S_1$ represents the earliest fabric observed within the mafic granulite, garnet amphibolite, biotite gneiss, and SBIFs. The SBIFs portray a dominant $S_2$ foliation, parallel to the $S_1$ gneissic compositional layering (Figure 3c). The $S_1/S_2$ in the northern prospect represents an NW shallow-dipping composite fabric with a mean direction of N126°E32NE, whereas, in the southern prospect, $S_2$ (Figure 3d), which affected SBIFs and biotite gneiss, is steeply dipping 60–75° towards the SW. The average direction of $S_1/S_2$ is N145°E66SW. Broad mesoscale gentle folds are observed on the SBIFs and biotite gneiss (Figure 3e). The C-planes of these fabrics are parallel to $S_1/S_2$, suggesting a component of layer parallel shearing (Figure 3f). The disposition of the foliation planes $S_1/S_2$ from the two prospects in the diagram of poles (Figure 2b) reflects the existence of a regional fold with axial plane oriented N049E and dipping 78° to the SE.

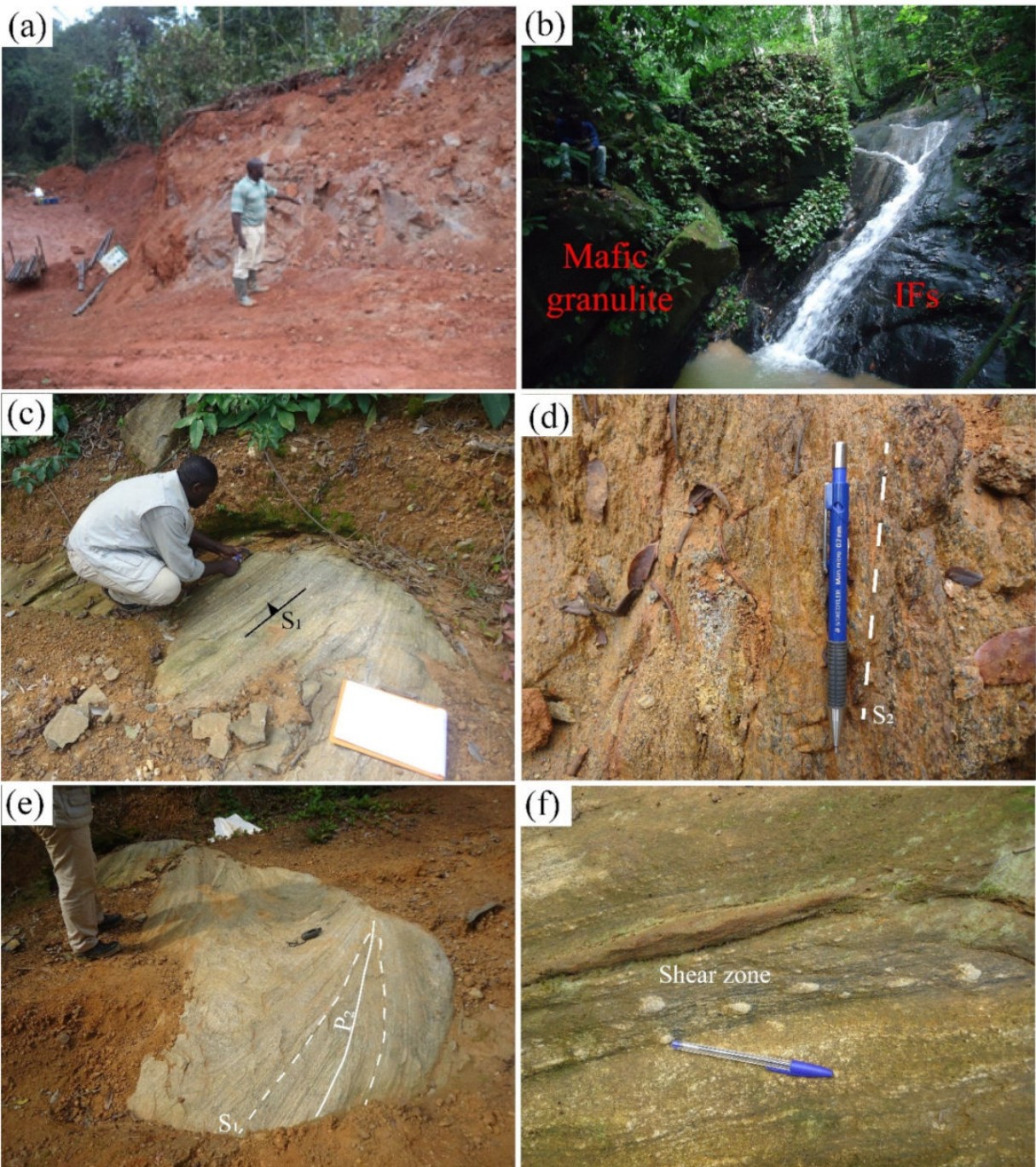

**Figure 3.** Outcrop views and deformation features: sheared banded iron formation (SBIF) outcropping as (**a**) road cuttings; (**b**) river bed generally associated with mafic granulite; (**c**) $S_1$ foliation in Biotite gneiss; (**d**) $S_2$ foliation in SBIF; (**e**) $P_2$ fold in biotite gneiss; (**f**) Shear zone in biotite gneiss.

## 3. Lithostratigraphy of the Anyouzok Iron Ore Deposit

The lithostratigraphy of the study area was determined via the logging of thirteen representative holes drilled by Caminex SARL. Eight drillholes (TH9, TD37, TD35, TD46, TH31, TD47, TD60, and TD65) were logged from the northern prospect (Figure 4a), while five drillholes (TE4, TE5, TE6, TE9, and TE22) were considered for the southern prospect (Figure 4b). The drillhole details are presented in Supplementary Tables S1 and S2. The units intercepted along the stratigraphy in both prospects are metamorphosed and consist of an IFs unit and country rocks unit (mafic granulite, garnet amphibolite, and biotite gneiss).

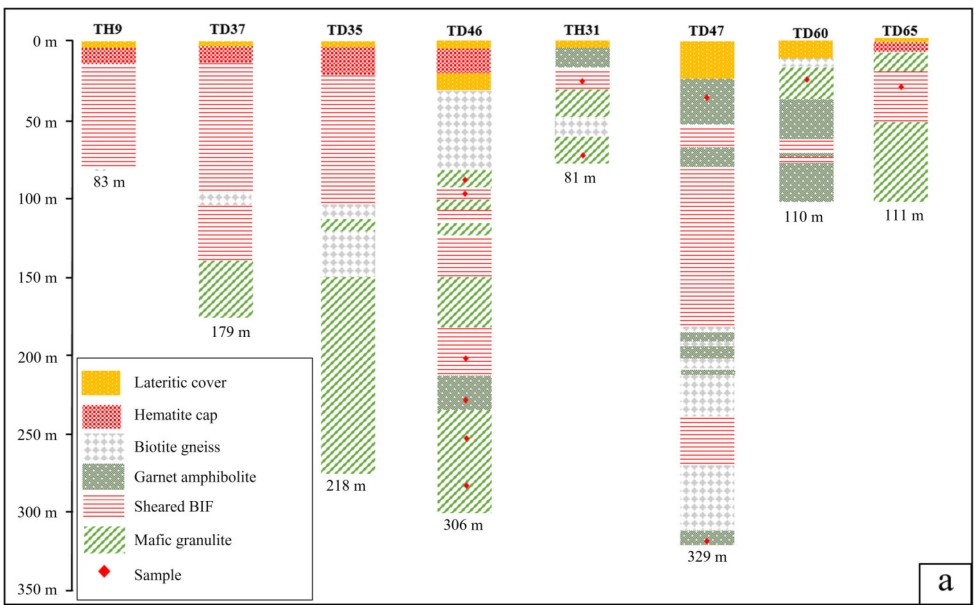

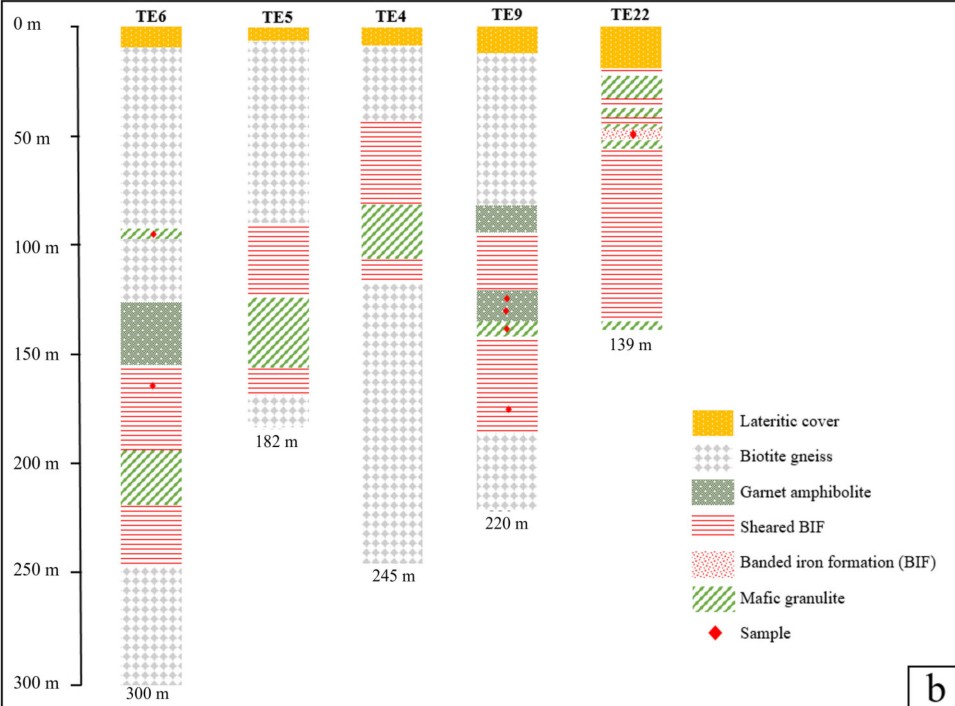

**Figure 4.** Stratigraphic logs with sample locations at the Anyouzok (**a**) northern prospect and (**b**) southern prospect.

### 3.1. The Northern Prospect

Iron formations unit

In this part of the deposit, IFs consist of SBIFs, intercepted below the surface at variable depths from 15 m (hole TD37), with cumulative thicknesses ranging from 10.30 m (hole TD60; Figure 4a) to 147.3 m (hole TD47; Figure 4a). SBIFs are found in sharp to gradational contact with interbedded biotite gneiss, mafic granulite, and garnet amphibolite. The core specimen consists mainly of medium- to coarse-grained magnetite and quartz.

Country rocks unit

Biotite gneiss is intercepted from a 12.6 m depth (hole TD60) with various intercalations along the section and a cumulative thickness up to 86.80 m (hole TD35). Biotite gneiss at the hanging wall was intercepted from ca. 32–84 m, just below the hematite cap (hole TD46; Figure 5b). At the footwall, biotite gneiss is found in sharp contact with mafic granulite and garnet amphibolite. The core specimen is mainly made up of fine- to coarse-grained biotite, quartz, and feldspar.

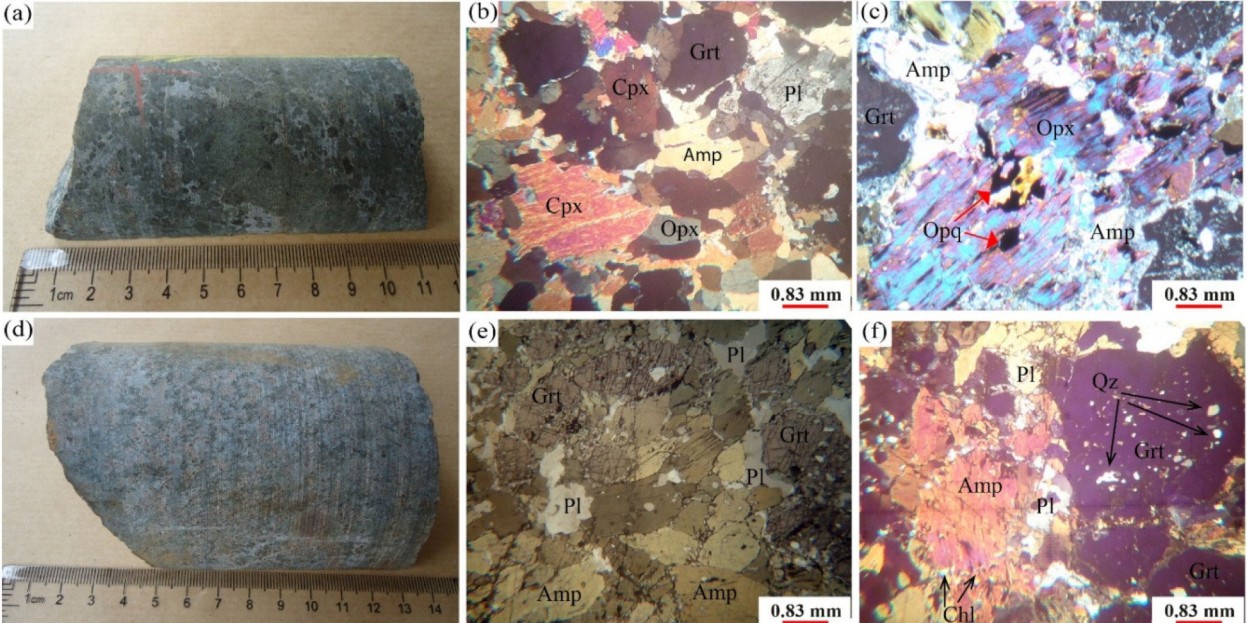

**Figure 5.** Drill core samples and photomicrographs of the Anyouzok metavolcanic rocks: (**a**) hand specimen of mafic granulite; (**b**) granoblastic heterogranular microstructure with pyroxene, hornblende, and garnet association (Plane polarized light: PPL); (**c**) anhedral orthopyroxene undergoing transformation into amphibole along the rims; (**d**) core sample of garnet amphibolite; (**e**) granoblastic heterogranular microstructure with amphibole, garnet, and quartz association (PPL); (**f**) amphibole grain boundaries partially replaced by chlorite (Cross Polarized Light: XPL). Abbreviations: Grt: garnet; Cpx: clinopyroxene; Opx: orthopyroxene; Amp: amphibole; Qz: quartz; Chl: chlorite; Pl: plagioclase.

Mafic granulite represents the major lithotype along the stratigraphy of the northern prospect (Figure 4a) and generally shows a sharp and conformable contact with the SBIFs and other country rocks. At the hanging wall, it is intercepted in holes TD46, TD60, and TH31, with thicknesses of 11.5 m, 23.1 m, and 13.36 m, respectively. The core specimen consists of medium- to coarse-grained pyroxene, garnet, amphibole, and quartz.

Garnet amphibolite represents the less abundant lithotype along the stratigraphy of the northern prospect (Figure 4a). It is intercepted from 3.60 m below the surface (hole TH31) to a 329.16 m depth (hole TD47), with a cumulative thickness ranging from 7.12 m (hole TD35) to 70.16 m (hole TD47). The hand specimen shows visible amphibole, garnet, and quartz crystals.

### 3.2. The Southern Prospect

Iron formation unit

The IFs unit within the southern prospect comprises ubiquitous SBIFs and minor BIFs, the latter only intercepted along the section of the drillhole TE22.

SBIFs are exposed in the drill core at various depths from 19.4 m (hole TE22) below the surface and show cumulative thicknesses ranging from 46.0 m (hole TE4) to 92.6 m (hole

TE22; Figure 4b). In this prospect, SBIFs mainly show sharp and conformable contacts with intercalated mafic granulite and garnet amphibolite. Likewise, on the northern prospect, the drill core specimen mainly consists of magnetite and quartz.

BIFs were not encountered during surface mapping but are exposed in the drill core of the hole TE22 from 46.5 to 48.0 m, with a thickness of 1.5 m. They occur sandwiched with gradational contacts between mafic granulite (Figure 4b). The core specimen is banded, with alternating white and dark millimeter- to centimeter-thick quartz-rich and magnetite-rich bands, respectively. Within the dark bands, the magnetite is fine- to medium-grained, while the white bands show fine-grained quartz.

Country rocks unit

Biotite gneiss is intercepted at various depths from 8.5 m (hole TE4) below the surface (Figure 4b) and shows cumulative thicknesses ranging from 1.51 m (hole TE22) to 165.95 m (hole TE4). It represents the most abundant lithology of the southern prospect (Figure 4b). Biotite gneiss mainly shows gradational contacts with interbedded mafic granulite, SBIFs, and garnet amphibolite (Supplementary Table S2). It is mainly made up of fine- to coarse-grained biotite, quartz, and feldspar crystals.

Mafic granulite mainly occurs as intercalations between SBIFs with generally sharp contacts and has cumulative thicknesses ranging from 6.25 m (Hole TE9) to 31.75 m (hole TE5). Along the section of hole TE22, mafic granulite represents the only lithology intercalated between SBIFs and BIFs. The core specimen consists of medium- to coarse-grained pyroxene, garnet, amphibole, and quartz.

Garnet amphibolite in the southern prospect (Figure 4b) as well as in the northern prospect represents the least abundant lithotype. The intercepts extend from 80.7 m (hole TE9) below the surface to 125 m (hole TE6), with cumulative thicknesses ranging from 28.25 m (Hole TE9) to 28.8 m (hole TE6). The drill core specimen shows medium to coarse amphibole, garnet, and quartz.

## 4. Sampling and Analytical Methods

Diamond drill cores were logged at the Caminex camp core shed. For each hole, drill cores were observed from the beginning to the end of the hole. Based on color, texture, and mineral composition, different rock units were distinguished and rock boundaries were delimited. Using a measuring tape, lithologic boundaries and thicknesses were measured and recorded progressively. Half and quarter core samples were systematically collected from the representative drillholes with respect to their lithology and texture. Drill core samples were carefully selected to ensure that the full variability of the iron mineralized unit (SBIFs and BIFs) and interbedded un-mineralized unit (barren zones) were represented, as they need to reflect the geology of the deposit. A total of 31 samples (8 SBIFs, 4 BIFs, 8 mafic granulites, and 11 garnet amphibolites) from 8 drillholes (TH31, TD46, TD60, TE6, TE9, TD47, TE22, and TD65) were collected for this study. Details of holes logged and/or sampled are presented in Supplementary Tables S1 and S2. Selected samples were placed in plastic bags and sealed up to prevent contamination. Labels and codes were given to the samples, which were written on the bags.

Twenty standard thin sections of metavolcanic rocks (ten for mafic granulite and ten for garnet amphibolite) and ten polished thin sections of IFs were prepared at Vanpetro and Geotech Lab, Vancouver (Canada). Detailed microscopic description was carried out at the Earth Sciences Department, University of Yaoundé I (Cameroon) and at Vanpetro Lab, Vancouver (Canada). The IFs' mineral composition was further determined by the X-ray diffraction (XRD) method with the Bruker D8-Advanced Eco 1Kw diffractometer (Bruker Corporation, Billerica, MA, USA) at AGES (University of Liège, Liège, Belgium). A total of 100 g of fresh rock sample were carefully selected, cut, manually crushed, and sieved to 250 μm, using a mortar and pestle. X-ray wavelengths X $(0.1 < 1 < 10\ \mu m)$ are incident on the mineral sample. Excited atoms emit radiations consistent with Bragg's law, $\lambda = 2d(hkl)\sin\theta$, where n = whole numbers corresponding to the order of diffraction; $\lambda$ = incident wavelength, d = distance between layers, and $\theta$ = angle of diffraction. Based

on the petrographic results, twenty-seven representative fresh samples (six mafic granulites, ten garnet amphibolites, seven SBIFs, and four BIFs) were selected for whole rock geochemical analysis.

Whole rock geochemical analysis for major elements was conducted using rock pulp by inductively coupled plasma-atomic emission spectrometry (ICP-AES), while inductively coupled plasma mass spectrometry (ICP-MS) was conducted for trace element and rare earth element (REE) analysis at ALS Lab, Tipperary, Ireland. The samples were initially pulverized, and 50–60 g were extracted for analysis. Rock powder (0.2 g) was then fused with $LiBO_2$ and dissolved in 100 $mm^3$ of 5% $HNO_3$. Analytical uncertainties vary from 0.1% to 0.04% for major elements, from 0.1 to 0.5 ppm for trace elements, and from 0.01 to 0.5 ppm for rare earth elements. Loss on ignition (LOI) was determined by weight difference after ignition at 1000 °C. Various standards were used, and data quality assurance was verified by running these standards between samples as unknowns. Analysis precision for rare earth elements is estimated at 5% for concentrations >10 ppm and 10% when lower. After data curation, 7 altered samples were screened out, and 20 samples (4 mafic granulites, 7 garnet amphibolites, 5 SBIFs, and 4 BIFs) were selected for geochemical studies. Since Y is more similar to Ho and has been extensively used in REE studies of aqueous solutions and their precipitates [52], it has been inserted between Dy and Ho. REE-Y concentrations of IFs were normalized to Post Archaean Australian Shale (PAAS; [53]). The Eu, Ce, La, Gd, Y, and Pr anomalies of IFs discussed in this study are calculated following the procedure of Bau and Dulski [54] and Bolhar et al. [6]:

$(Eu/Eu^*)_{SN} = (Eu)_{SN}/(0.67Sm_{SN} + 0.33Tb_{SN})$; $(Ce/Ce^*)_{SN} = Ce_{SN}/(0.5La_{SN} + 0.5Pr_{SN})$; $(La/La^*)_{SN} = (La)_{SN}/(3Pr_{SN} - 2Nd_{SN})$; $(Gd/Gd^*)_{SN} = (Gd)_{SN}/(0.33Sm_{SN} + 0.67Tb_{SN})$; $(Y/Y^*)_{SN} = 2Y_{SN}/(Dy_{SN} + Ho_{SN})$; $(Pr/Pr^*)_{SN} = Pr_{SN}/(0.5Ce_{SN} + 0.5Nd_{SN})$.

## 5. Results

### 5.1. Petrography and Mineralogy

#### 5.1.1. Metavolcanic Rocks

Metavolcanic rocks appear as mafic granulite and garnet amphibolite.

Mafic granulite tends to be massive and generally dark grey, with coarse-grained pyroxene associated with garnet and patchy amphibole crystals (Figure 5a). In thin sections, the rock shows a granoblastic heterogranular microstructure (Figure 5b), consisting of pyroxene (35 vol.%), garnet (30 vol.%), amphibole (10 vol.%), plagioclase (10 vol.%), quartz (5 vol.%), opaque (2 vol.%), and chlorite (<2 vol.%). Pyroxene occurs mainly as clinopyroxene and orthopyroxene in lesser volumes. They are medium- to coarse-grained (up to 2 mm in size). They show transformation into amphibole and opaque minerals along their rims and cleavages (Figure 5c). Garnet is medium- to coarse-grained (up to 2 mm), sub-angular to rounded, and fractured. It sometimes contains quartz inclusions (Figure 5b,c). Quartz is fine- to coarse-grained and generally appears either as aggregates mixed with opaque minerals or as occupying interstices between minerals. Amphibole is medium- to coarse-grained (0.5–2 mm) and occurs generally as angular to sub-angular crystals, associated with garnet, clinopyroxene, and quartz. It shows transformation into chlorite. Plagioclase is anhedral, fine- to medium-grained (up to 2 mm), and shows transformation into sericite (Figure 5b). Opaque minerals are rare and found as inclusions within pyroxene, garnet, and amphibole.

Garnet amphibolite is generally medium- to coarse-grained and brownish-gray in color, with coarse-grained garnet surrounded by amphibole (Figure 5d). In thin sections, the rock shows a granoblastic heterogranular microstructure (Figure 5e) consisting of amphibole (50 vol.%), garnet (15 vol.%), plagioclase (25 vol.%), pyroxene (5 vol.%) quartz (<2 vol.%), and opaque (<3 vol.%). Amphibole is medium- to coarse-grained (1–2 mm) and occurs as rounded to sub-angular crystals. It sometimes shows transformation into chlorite and opaque minerals along the rims and cleavages (Figure 5f). Garnet is generally sub-rounded to polygonal, medium- to coarse-grained (1–2 mm in size), and fractured. Plagioclase is fine- to medium-grained (<0.5 mm) and occurs as a crushed mosaic crystal,

in association with garnet, hornblende, and pyroxene. Minute quartz inclusions are also found in garnet and hornblende (Figure 5f). Pyroxene occurs as medium-grained (~1 mm) and angular to sub-rounded crystals. Opaque minerals are fine-grained and appear as inclusions in garnet and amphibole crystals or along their rims (Figure 5e).

### 5.1.2. Iron Formations

The Anyouzok IFs comprise banded iron formations (BIFs) and sheared banded iron formations (SBIFs).

BIF is fine- to medium-grained, foliated, and made up of alternating white silica-rich bands and dark magnetite-rich bands (Figure 6a). The white bands range from 1 to 10 mm in thickness, while the dark bands range from 2 to 16 mm in thickness. In thin sections, the rock has a granoblastic heterogranular microstructure (Figure 6b), consisting of quartz (50 vol.%), magnetite (30 vol.%), amphibole (15 vol.%), pyrite (2 vol.%), biotite (1 vol.%), chlorite (<1 vol.%), and hematite (<1 vol.%). Magnetite is fine-grained (0.5 × 0.5 mm on average), anhedral, and in association with amphibole and biotite. It sometimes appears intergrown with pyrite and, less likely, with hematite (Figure 6c). Some magnetite crystals contain minute quartz inclusions. Amphibole crystals are generally stretched and occur as tremolite and actinolite of 1.5 mm, on average (Figure 6b). They are mostly found partly altered into chlorite and contain diffuse magnetite inclusions (Figure 6b). Biotite (0.25 × 0.5 mm) is anhedral and generally stretched, and it is closely associated with amphibole, magnetite, and, rarely, quartz. It sometimes presents transformation into chlorite, mostly along the rims. Quartz crystals are anhedral and in close association with amphibole, magnetite, and biotite. Pyrite is fine- to medium-grained and anhedral to subhedral. Hematite is rare and occurs as traces at intimate intergrowths of pyrite with magnetite (Figure 6c).

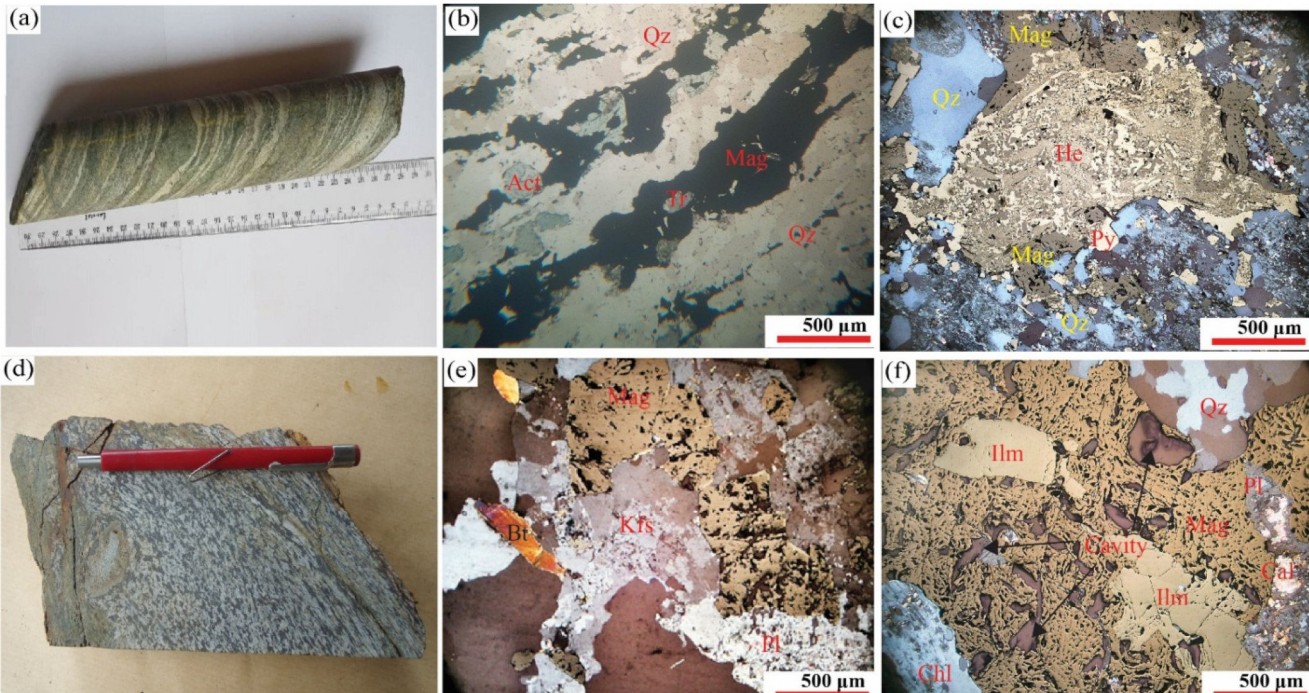

**Figure 6.** Drill core and photomicrographs (partially crossed and reflected light) of the Anyouzok IFs: (**a**) hand specimen of Banded iron formation (BIF) showing alternating silica and magnetite-rich bands; (**b**) photomicrograph of BIF showing granoblastic heterogranular microstructure; (**c**) anhedral magnetite crystal partially replaced by hematite and pyrite; (**d**) core specimen of SBIF; (**e**) granoblastic magnetite associated with biotite and intensely sericitized plagioclase and K-feldspar; (**f**) anhedral magnetite exhibiting several cavities, in association with ilmenite and secondary calcite. Abbreviations: Act: actinolite; Mag: magnetite; Bt: biotite; Ilm: Ilmenite; Qz: Quartz; Cal: Calcite; Pl: Plagioclase; Kfs: K-feldspar; Py: pyrite; Chl: chlorite.

SBIF is dark grey, medium- to coarse-grained, and massive to highly strained; it is mainly composed of magnetite and quartz (Figure 6d). In thin sections, the rock shows a mylonitic and granoblastic heterogranular microstructure (Figure 6e), consisting of magnetite (60 vol.%), quartz (20 vol.%), amphibole (10 vol.%), plagioclase (2 vol.%), ilmenite (2 vol.%), biotite (2 vol.%), chlorite (3 vol.%), calcite (1 vol.%), and K-feldspar (<1 vol.%). Magnetite is found as anhedral and coarse crystals, though very few euhedral grains are also present (Figure 6e). It contains several cavities and is occasionally associated with ilmenite (Figure 6f). Where magnetite crystals lack cavities, quartz is found as minute inclusions in magnetite. Quartz is anhedral, generally coarse-grained, and associated with magnetite and amphibole. Plagioclase and K-feldspar are anhedral, medium- to coarse-grained, and generally altered to sericite. Amphibole occurs as tremolite and actinolite. It is anhedral, coarse-grained (1 × 1.5 mm), and generally altered into chlorite (Figure 6f). Biotite is anhedral, fine- to medium-grained, and associated with plagioclase, K-feldspar, quartz, and, to a lesser extent, magnetite. Calcite is anhedral and is generally found at the rims of magnetite crystals (Figure 6f).

### 5.1.3. Mineralogy of IFs

The mineralogy of the Anyouzok BIFs investigated by the XRD method (Figure 7a) is simple, with main mineral peaks consisting of magnetite (2.53 Å), hematite (1.49 Å), quartz (2.34 Å), biotite (10.10 Å), pyrite (2.42 Å), and tremolite (6.37 Å). Like the BIFs, the SBIFs (Figure 7b) present a similar mineralogy, with peaks of magnetite (2.53 Å), hematite (1.49 Å), quartz (3.34 Å), biotite (10.10 Å), pyrite (2.42 Å), and tremolite (2.70 Å). In addition, peaks of secondary minerals not detected in BIF, such as calcite (2.29 Å) and chlorite (7.07 Å), are also detected.

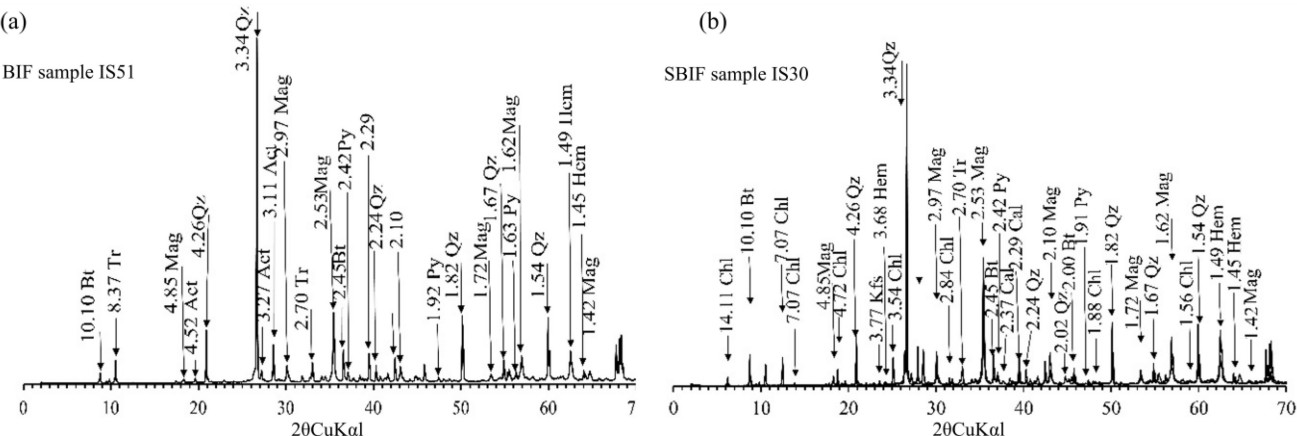

**Figure 7.** X-ray diffraction spectra for the Anyouzok BIF (**a**) and SBIF (**b**) samples, indicating prominent quartz and magnetite peaks. Magnetite (Mag), biotite (Bt), hematite (Hem), quartz (Qz), chlorite (Chl), tremolite (Tr), actinolite (Act), pyrite (Py), Calcite (Cal).

### 5.2. Geochemistry

Whole rock geochemical compositions of the Anyouzok metavolcanic rocks and interbedded IFs are presented in Tables 1 and 2, respectively.

**Table 1.** Major elements (wt%) and trace elements (ppm) compositions of the Anyouzok metavolcanic rocks.

| Rock Type | Mafic Granulite | | | | | Garnet Amphibolite | | | | | |
|---|---|---|---|---|---|---|---|---|---|---|---|
| Major Elements | IS20 | IS40 | IS26 | IS17 | IS15b | IS24 | IS27b | IS28 | IS33 | IS37 | IS39 |
| $SiO_2$ | 44.70 | 48.80 | 50.60 | 51.20 | 44.10 | 44.80 | 49.30 | 46.50 | 49.10 | 50.80 | 46.50 |
| $TiO_2$ | 0.57 | 0.90 | 2.24 | 1.77 | 1.84 | 0.64 | 1.37 | 1.46 | 1.34 | 0.99 | 0.77 |
| $Al_2O_3$ | 14.25 | 14.75 | 11.95 | 13.50 | 12.90 | 14.05 | 13.20 | 14.35 | 12.05 | 13.30 | 13.65 |

Table 1. *Cont.*

| Rock Type | Mafic Granulite | | | | | Garnet Amphibolite | | | | | |
|---|---|---|---|---|---|---|---|---|---|---|---|
| Major Elements | IS20 | IS40 | IS26 | IS17 | IS15b | IS24 | IS27b | IS28 | IS33 | IS37 | IS39 |
| $Fe_2O_3$ | 14.65 | 16.60 | 21.50 | 18.15 | 18.10 | 12.25 | 17.45 | 15.80 | 18.60 | 16.45 | 14.25 |
| MgO | 9.91 | 5.92 | 4.01 | 5.07 | 7.72 | 11.30 | 6.28 | 8.99 | 5.50 | 6.37 | 11.40 |
| MnO | 0.21 | 0.27 | 0.25 | 0.22 | 0.26 | 0.17 | 0.21 | 0.20 | 0.22 | 0.21 | 0.20 |
| CaO | 13.95 | 8.28 | 7.82 | 8.14 | 11.85 | 13.25 | 9.31 | 7.73 | 9.37 | 8.65 | 9.33 |
| $Na_2O$ | 1.70 | 1.48 | 0.41 | 1.16 | 1.25 | 1.70 | 1.06 | 2.04 | 1.58 | 1.15 | 1.73 |
| $K_2O$ | 0.29 | 0.55 | 0.07 | 0.48 | 0.53 | 0.54 | 0.43 | 0.75 | 0.45 | 0.57 | 0.91 |
| $Cr_2O_3$ | 0.07 | 0.01 | <0.01 | 0.01 | 0.02 | 0.06 | 0.01 | 0.02 | 0.02 | 0.02 | 0.12 |
| $P_2O_5$ | 0.05 | 0.32 | 0.25 | 0.22 | 0.30 | 0.05 | 0.13 | 0.18 | 0.19 | 0.17 | 0.06 |
| LOI | 0.04 | 1.02 | −0.97 | 0.05 | 0.38 | 0.49 | −0.15 | 2.21 | 0.17 | 0.41 | 1.71 |
| Total | 100.39 | 98.92 | 98.13 | 99.98 | 99.26 | 99.30 | 98.61 | 100.24 | 98.60 | 99.10 | 100.65 |
| mg# | 57.27 | 41.40 | 26.98 | 35.62 | 45.80 | 64.63 | 41.62 | 52.99 | 36.94 | 43.41 | 61.31 |
| $K_2O/Na_2O$ | 0.17 | 0.37 | 0.17 | 0.41 | 0.42 | 0.32 | 0.41 | 0.37 | 0.28 | 0.50 | 0.53 |
| Trace and rare earth elements | | | | | | | | | | | |
| Cr | 460.00 | 100.00 | 20.00 | 80.00 | 150.00 | 440.00 | 80.00 | 120.00 | 130.00 | 100.00 | 820.00 |
| Sn | 1.00 | 1.00 | 2.00 | 1.00 | 1.00 | 1.00 | 1.00 | 1.00 | 1.00 | 1.00 | 1.00 |
| V | 361.00 | 287.00 | 392.00 | 391.00 | 475.00 | 258.00 | 429.00 | 392.00 | 242.00 | 338.00 | 231.00 |
| Ba | 24.10 | 82.60 | 6.00 | 103.50 | 90.80 | 31.60 | 63.60 | 126.50 | 52.80 | 53.80 | 70.80 |
| Rb | 2.50 | 7.30 | 1.20 | 15.00 | 8.90 | 2.90 | 7.50 | 10.00 | 9.10 | 5.30 | 10.60 |
| Ga | 16.80 | 13.70 | 18.70 | 15.00 | 18.80 | 13.30 | 16.90 | 18.40 | 18.70 | 14.10 | 16.30 |
| Cs | 0.02 | 0.17 | 0.05 | 0.65 | 0.44 | 0.01 | 0.30 | 0.18 | 0.12 | 0.07 | 0.31 |
| Th | 0.15 | 0.93 | 1.33 | 1.64 | <0.5 | 0.07 | 0.40 | 0.16 | 0.69 | 0.50 | 0.44 |
| U | 0.10 | 0.51 | 0.31 | 2.63 | 0.24 | 0.08 | 0.24 | 0.27 | 0.27 | 0.35 | 0.31 |
| Sr | 48.40 | 86.40 | 20.70 | 62.10 | 69.50 | 37.50 | 32.30 | 53.10 | 57.00 | 38.60 | 40.70 |
| Nb | 0.60 | 3.70 | 6.90 | 6.60 | 6.60 | 1.30 | 4.70 | 4.70 | 6.30 | 3.10 | 3.00 |
| Ta | <0.1 | 0.30 | 0.30 | 0.30 | 0.30 | <0.1 | 0.30 | 0.20 | 9.70 | <0.1 | 0.10 |
| Hf | 1.30 | 1.80 | 4.80 | 3.80 | 3.40 | 1.10 | 2.60 | 2.80 | 3.80 | 2.40 | 2.50 |
| Zr | 37.00 | 76.00 | 176.00 | 140.00 | 115.00 | 34.00 | 91.00 | 103.00 | 125.00 | 87.00 | 94.00 |
| Y | 16.70 | 36.10 | 53.10 | 39.20 | 34.20 | 16.20 | 31.50 | 35.10 | 37.80 | 29.40 | 22.50 |
| La | 2.10 | 7.00 | 9.80 | 8.70 | 6.60 | 2.10 | 4.60 | 7.30 | 6.60 | 5.10 | 4.00 |
| Ce | 6.80 | 16.30 | 26.10 | 20.90 | 19.50 | 6.00 | 13.20 | 21.30 | 18.10 | 13.40 | 10.00 |
| Pr | 1.19 | 2.43 | 3.51 | 2.82 | 2.97 | 0.93 | 2.09 | 3.24 | 2.68 | 1.87 | 1.54 |
| Nd | 7.30 | 11.30 | 18.00 | 14.60 | 15.00 | 4.80 | 10.80 | 17.10 | 14.50 | 9.20 | 7.10 |
| Sm | 2.58 | 3.19 | 5.78 | 4.35 | 4.37 | 1.63 | 3.77 | 5.19 | 4.04 | 2.80 | 2.19 |
| Eu | 0.89 | 1.14 | 1.89 | 1.59 | 1.54 | 0.69 | 1.49 | 1.50 | 1.56 | 0.99 | 0.64 |
| Gd | 2.62 | 4.07 | 8.14 | 6.14 | 5.11 | 2.30 | 4.97 | 5.63 | 5.99 | 3.69 | 2.72 |
| Tb | 0.43 | 0.81 | 1.48 | 1.07 | 0.88 | 0.41 | 0.84 | 0.93 | 1.03 | 0.69 | 0.56 |
| Dy | 3.14 | 5.49 | 9.93 | 7.80 | 6.44 | 2.86 | 5.74 | 6.33 | 6.88 | 5.20 | 3.66 |
| Ho | 0.64 | 1.29 | 2.06 | 1.49 | 1.29 | 0.63 | 1.27 | 1.30 | 1.42 | 1.15 | 0.76 |
| Er | 1.98 | 3.84 | 6.28 | 4.50 | 4.08 | 2.17 | 3.77 | 3.77 | 4.45 | 3.71 | 2.44 |
| Tm | 0.24 | 0.59 | 0.77 | 0.62 | 0.50 | 0.26 | 0.50 | 0.49 | 0.56 | 0.42 | 0.34 |
| Yb | 1.85 | 4.01 | 6.15 | 4.24 | 3.99 | 2.03 | 3.58 | 3.84 | 4.54 | 3.45 | 2.89 |
| Lu | 0.25 | 0.65 | 0.85 | 0.63 | 0.56 | 0.29 | 0.52 | 0.52 | 0.60 | 0.51 | 0.47 |
| ΣREE | 32.01 | 62.11 | 100.74 | 79.45 | 72.83 | 27.10 | 57.14 | 78.44 | 72.95 | 52.18 | 39.31 |
| $(La/Yb)_{CN}$ | 0.77 | 1.19 | 1.08 | 1.39 | 1.12 | 0.70 | 0.87 | 1.29 | 0.99 | 1.00 | 0.94 |
| $(La/Sm)_{CN}$ | 0.51 | 1.37 | 1.06 | 1.25 | 0.94 | 0.80 | 0.76 | 0.88 | 1.02 | 1.14 | 1.14 |
| $(Gd/Yb)_{CN}$ | 1.15 | 0.82 | 1.07 | 1.17 | 1.04 | 0.92 | 1.12 | 1.19 | 1.07 | 0.87 | 0.76 |
| $(Eu/Eu*)_{CN}$ | 1.04 | 0.96 | 0.84 | 0.94 | 0.99 | 1.09 | 1.05 | 0.85 | 0.97 | 0.94 | 0.80 |
| $(Ce/Ce*)_{CN}$ | 1.04 | 0.96 | 1.08 | 1.02 | 1.07 | 1.04 | 1.03 | 1.06 | 1.04 | 1.05 | 0.97 |
| Th/Nb | 0.25 | 0.25 | 0.19 | 0.25 | 0.00 | 0.05 | 0.09 | 0.03 | 0.11 | 0.16 | 0.15 |
| Nb/Y | 0.04 | 0.10 | 0.13 | 0.17 | 0.19 | 0.08 | 0.15 | 0.13 | 0.17 | 0.11 | 0.13 |
| La/Nb | 3.50 | 1.89 | 1.42 | 1.32 | 1.00 | 1.62 | 0.98 | 1.55 | 1.05 | 1.65 | 1.33 |
| Dy/Yb | 1.70 | 1.37 | 1.61 | 1.84 | 1.61 | 1.41 | 1.60 | 1.65 | 1.52 | 1.51 | 1.27 |
| Zr/Nb | 61.67 | 20.54 | 25.51 | 21.21 | 17.42 | 26.15 | 19.36 | 21.91 | 19.84 | 28.06 | 31.33 |
| Zr/Hf | 28.46 | 42.22 | 36.67 | 36.84 | 33.82 | 30.91 | 35.00 | 36.79 | 32.89 | 36.25 | 37.60 |
| Th/Yb | 0.08 | 0.23 | 0.22 | 0.39 | 0.01 | 0.03 | 0.11 | 0.04 | 0.15 | 0.14 | 0.15 |

**Table 2.** Major elements (wt%) and trace elements (ppm) compositions of the Anyouzok Iron formations.

| Rock Type | BIF | | | | SBIF | | | | |
|---|---|---|---|---|---|---|---|---|---|
| **Major Elements** | **IS13** | **IS50** | **IS51** | **IS52** | **IS18** | **IS19** | **IS35** | **IS41** | **IS54** |
| $SiO_2$ | 44.40 | 60.78 | 59.02 | 59.45 | 43.70 | 43.50 | 43.30 | 45.70 | 47.26 |
| $TiO_2$ | 0.12 | 0.07 | 0.07 | 0.05 | 0.01 | 0.04 | 0.03 | 0.08 | <0.01 |
| $Al_2O_3$ | 1.83 | 2.37 | 2.52 | 1.84 | 0.26 | 0.94 | 0.70 | 0.99 | 0.74 |
| $Fe_2O_3$ | 48.40 | 26.37 | 26.05 | 27.68 | 55.20 | 51.30 | 53.20 | 50.20 | 49.60 |
| $MgO$ | 2.55 | 5.27 | 5.98 | 5.73 | 1.94 | 1.94 | 2.08 | 2.21 | 1.82 |
| $MnO$ | 0.04 | 0.40 | 0.46 | 0.46 | 0.03 | 0.04 | 0.03 | 0.04 | 0.05 |
| $CaO$ | 1.46 | 2.44 | 3.39 | 2.76 | 1.16 | 1.10 | 1.03 | 0.75 | 1.10 |
| $Na_2O$ | 0.52 | 0.14 | 0.18 | 0.10 | 0.04 | 0.27 | 0.10 | 0.01 | 0.20 |
| $K_2O$ | 0.47 | 1.16 | 1.20 | 0.84 | 0.06 | 0.40 | 0.08 | 0.03 | 0.32 |
| $Cr_2O_3$ | 0.01 | 0.00 | 0.00 | 0.00 | <0.01 | 0.01 | 0.01 | 0.01 | 0.00 |
| $P_2O_5$ | 0.12 | 0.09 | 0.07 | 0.09 | 0.09 | 0.16 | 0.14 | 0.11 | 0.10 |
| LOI | −0.69 | 0.80 | 0.90 | 0.90 | −1.32 | −1.20 | −0.88 | −0.58 | −1.30 |
| Total | 99.24 | 99.88 | 99.87 | 99.87 | 101.17 | 98.51 | 99.82 | 99.56 | 99.96 |
| Fe | 33.88 | 18.46 | 18.24 | 19.38 | 38.64 | 35.91 | 37.24 | 35.14 | 34.72 |
| Fe/Si | 1.63 | 0.65 | 0.66 | 0.70 | 1.89 | 1.77 | 1.84 | 1.65 | 1.57 |
| Trace and rare earth elements | | | | | | | | | |
| Sn | <1 | <1 | <1 | <1 | <1 | <1 | <1 | <1 | <1 |
| V | 28.00 | <8 | <8 | <8 | 6.00 | 13.00 | 9.00 | 32.00 | <8 |
| Ba | 66.10 | 71.00 | 74.00 | 39.00 | 9.70 | 134.00 | 5.10 | 3.80 | 74.00 |
| Rb | 23.60 | 85.40 | 88.00 | 54.50 | 2.80 | 33.70 | 3.40 | 1.90 | 15.90 |
| Ga | 3.50 | 1.50 | 2.40 | 1.40 | 0.50 | 1.90 | 1.00 | 3.40 | 3.60 |
| Cs | 0.77 | 8.70 | 8.70 | 4.40 | 0.07 | 0.88 | 0.10 | 0.14 | 0.40 |
| Th | 1.23 | 1.50 | 1.30 | 1.10 | 0.05 | <0.05 | 0.13 | 0.16 | <0.2 |
| U | 0.32 | 0.30 | 0.30 | 0.20 | 0.10 | 0.08 | 0.38 | 0.09 | 0.10 |
| Sr | 10.50 | 8.90 | 11.50 | 6.60 | 7.10 | 20.70 | 5.00 | 5.30 | 15.00 |
| Nb | 1.00 | 0.50 | 0.60 | 0.40 | 0.20 | 0.40 | 1.00 | 0.70 | <0.1 |
| Ta | <0.1 | <0.1 | <0.1 | <0.1 | <0.1 | <0.1 | 0.10 | <0.1 | <0.1 |
| Hf | 0.50 | 0.40 | 0.50 | 0.30 | 0.20 | 0.20 | 0.20 | 0.20 | 0.10 |
| Zr | 18.00 | 17.00 | 16.90 | 12.40 | 4.00 | 7.00 | 7.00 | 5.00 | 3.70 |
| Y | 9.10 | 6.60 | 7.50 | 6.80 | 5.80 | 7.20 | 5.90 | 5.50 | 6.90 |
| La | 6.00 | 6.80 | 6.50 | 6.20 | 1.70 | 2.50 | 1.40 | 3.70 | 3.20 |
| Ce | 11.40 | 12.70 | 12.50 | 12.00 | 3.50 | 5.90 | 3.50 | 7.20 | 5.30 |
| Pr | 1.29 | 1.36 | 1.31 | 1.36 | 0.43 | 0.74 | 0.44 | 0.92 | 0.62 |
| Nd | 5.00 | 5.30 | 5.20 | 5.40 | 1.80 | 3.10 | 2.30 | 3.10 | 2.70 |
| Sm | 1.15 | 1.00 | 1.08 | 0.98 | 0.44 | 0.71 | 0.54 | 0.84 | 0.54 |
| Eu | 0.66 | 0.45 | 0.45 | 0.45 | 0.36 | 0.47 | 0.36 | 0.38 | 0.35 |
| Gd | 1.27 | 1.19 | 1.16 | 1.10 | 0.70 | 0.95 | 0.74 | 0.69 | 0.87 |
| Tb | 0.17 | 0.18 | 0.18 | 0.17 | 0.10 | 0.13 | 0.11 | 0.12 | 0.12 |
| Dy | 1.43 | 1.05 | 1.04 | 1.14 | 0.86 | 1.02 | 0.89 | 0.75 | 0.76 |
| Ho | 0.30 | 0.25 | 0.26 | 0.24 | 0.20 | 0.22 | 0.17 | 0.20 | 0.18 |
| Er | 0.93 | 0.74 | 0.79 | 0.70 | 0.59 | 0.64 | 0.56 | 0.43 | 0.55 |
| Tm | 0.09 | 0.10 | 0.11 | 0.10 | 0.05 | 0.05 | 0.05 | 0.07 | 0.08 |
| Yb | 0.81 | 0.74 | 0.82 | 0.68 | 0.66 | 0.66 | 0.56 | 0.41 | 0.51 |
| Lu | 0.09 | 0.11 | 0.13 | 0.11 | 0.06 | 0.04 | 0.04 | 0.09 | 0.08 |
| $\sum$REE-Y | 39.69 | 38.57 | 39.03 | 37.43 | 17.25 | 24.33 | 17.56 | 24.40 | 22.76 |
| $(Eu/Eu^*)_{SN}$ | 2.68 | 1.94 | 1.86 | 2.01 | 3.17 | 2.83 | 2.72 | 2.14 | 2.54 |
| $(Ce/Ce^*)_{SN}$ | 0.94 | 0.96 | 0.99 | 0.95 | 0.94 | 0.99 | 1.02 | 0.90 | 0.86 |
| $(La/La^*)_{SN}$ | 1.10 | 1.19 | 1.23 | 1.13 | 1.12 | 0.96 | 2.66 | 0.75 | 1.63 |
| $(Gd/Gd^*)_{SN}$ | 1.35 | 1.26 | 1.21 | 1.22 | 1.41 | 1.40 | 1.32 | 1.03 | 1.45 |
| $(Y/Y^*)_{SN}$ | 0.86 | 0.80 | 0.89 | 0.81 | 0.87 | 0.94 | 0.94 | 0.88 | 1.16 |
| $(Pr/Pr^*)_{SN}$ | 1.01 | 0.98 | 0.96 | 0.99 | 1.00 | 1.01 | 0.89 | 1.15 | 0.96 |
| $(Eu/Eu^*)_{CN}$ | 1.66 | 1.26 | 1.23 | 1.32 | 1.98 | 1.74 | 1.74 | 1.52 | 1.56 |
| $(Eu/Eu^*)_{NASC}$ | 2.70 | 1.97 | 1.88 | 2.04 | 3.26 | 2.88 | 2.78 | 2.16 | 2.61 |
| Y/Ho | 30.33 | 26.40 | 28.85 | 28.33 | 29.00 | 32.73 | 34.71 | 27.50 | 38.33 |
| Pr/Yb | 1.59 | 1.84 | 1.60 | 2.00 | 0.65 | 1.12 | 0.79 | 2.24 | 1.22 |
| Th/U | 3.84 | 5.00 | 4.33 | 5.50 | 0.50 | 0.31 | 0.34 | 1.78 | 1.00 |

### 5.2.1. Metavolcanic Rocks

Major elements

The metavolcanic samples present wide compositional variations in their major elements (Table 1), such as $SiO_2$ (44.7–1.2 wt%; 44.1–50.8 wt%), $Fe_2O_3$ (14.65–21.5 wt%; 12.25–18.6 wt%), MgO (4.01–9.91 wt%; 5.5–11.4 wt%), and CaO (7.82–13.95 wt%; 7.73–13.25 wt%) for mafic granulite and garnet amphibolite, respectively. The analyzed samples show low to moderate contents in $Al_2O_3$ (11.95–14.75 wt%; 12.05–14.35 wt%), $TiO_2$ (0.57–2.24 wt%; 0.64–1.84 wt%), MnO (0.21–0.27 wt%; 0.17–0.26 wt%), $P_2O_5$ (0.05–0.32 wt%; 0.05–0.30 wt%), and $Cr_2O_3$ (<0.01–0.07 wt%; 0.01–0.12 wt%). $Na_2O$ (0.41–1.7 wt%; 1.15–2.04 wt%) contents are higher than $K_2O$ (0.07–0.55 wt%; 0.43–0.91 wt%) contents in both rock types, yielding to low $K_2O/Na_2O$ ratios (0.17–0.41; 0.28–0.53). LOI values range from −0.97 to 1.02 wt% and −0.15 to 2.21 wt% for mafic granulite and garnet amphibolite, respectively.

Trace and rare earth elements (REE)

Except for Zr (37–176 ppm; 34–125 ppm) and Y (16.7–53.1 ppm; 16.2–37.8 ppm), the high field strength elements (HFSEs) in mafic granulite and garnet amphibolite samples are generally low, <10 ppm (Table 1). Large ion lithophile elements (LILEs) such as Rb (1.2–15 ppm; 2.9–10.6 ppm) show very low concentrations, whereas Sr (20.7–86.4 ppm; 32.3–69.5 ppm) and Ba (6–103.5 ppm; 31.6–126.5 ppm) show slightly higher values. Cr contents range from 20 to 460 ppm and from 80 to 820 ppm in mafic granulite and garnet amphibolite, respectively.

REE contents are variable and higher in mafic granulite samples ($\sum$REE: 32.01–100.74 ppm) compared to garnet amphibolite ($\sum$REE: 27.1–78.44 ppm). Chondrite-normalized [55] REE diagrams show homogeneous and coherent patterns for most of the analyzed samples, comparable to both NMORB and EMORB (Figure 8a,c). The Nyong Complex mafic granulites from Kribi [31] and Bipindi [11] and the Nyong Complex garnet amphibolites from Akom II [22] are plotted for comparison. Both the Anyouzok mafic granulite and garnet amphibolite samples, as well as the Nyong Complex mafic granulite samples, show relatively flat patterns ($(La/Yb)_{CN}$ = 0.77–1.39; 0.70–1.29 for mafic granulite and garnet amphibolite, respectively) (Figure 8a,c). In contrast, the Nyong Complex garnet amphibolite patterns are more fractionated, with LREE enriched over the HREE. The Anyouzok samples generally present slightly negative to no Eu anomalies ($(Eu/Eu^*)_{CN}$ = 0.84–1.04; 0.80–1.09) and exhibit no Ce anomalies (Ce/Ce*: 0.96–1.08; 0.97–1.07) compared to the Nyong Complex mafic granulite and garnet amphibolite. Primitive mantle-normalized [52] multi-element diagrams (Figure 8b,d) show peaks in K and U and troughs in Nb, Ta, Sr, and Ti for mafic granulite samples and in Th, Nb, Ta, and Sr for garnet amphibolite samples.

### 5.2.2. Iron Formations

Major elements

The major element compositions of the Anyouzok IFs show that $SiO_2$ and $Fe_2O_3$ are the main constituents, representing ca 96 wt% of the bulk composition of SBIFs and ca 87 wt% of BIFs (Table 2). BIFs show higher $SiO_2$ (44.4–60.78 wt%) and lower $Fe_2O_3$ (26.05–48.40 wt%) contents, while SBIFs show lower $SiO_2$ (43.3–47.26 wt%) and higher $Fe_2O_3$ (49.6–55.2 wt%) contents. Likewise, MgO (2.55–5.98 wt.%), CaO (1.46–3.39 wt%), $Al_2O_3$ (1.83–2.52 wt%), MnO (0.04–0.46 wt%), $TiO_2$ (0.05–0.12 wt%), and $K_2O$ (0.47–1.2 wt%) are of higher contents in BIFs than in SBIFs (MgO: 1.82–2.21 wt%; CaO: 0.75–1.16 wt%; $Al_2O_3$: 0.26–0.99 wt%; MnO: 0.03–0.05 wt%; $TiO_2$: 0.01–0.08 wt%; $K_2O$: 0.03–0.08 wt%). $P_2O_5$ concentrations have a narrow range for both IF types, ranging from 0.07–0.12 wt% and from 0.09–0.16 wt% for BIFs and SBIFs, respectively. LOI values range from −0.69 to 0.9 wt% in BIFs and from −1.32 to −0.58 wt% in SBIFs.

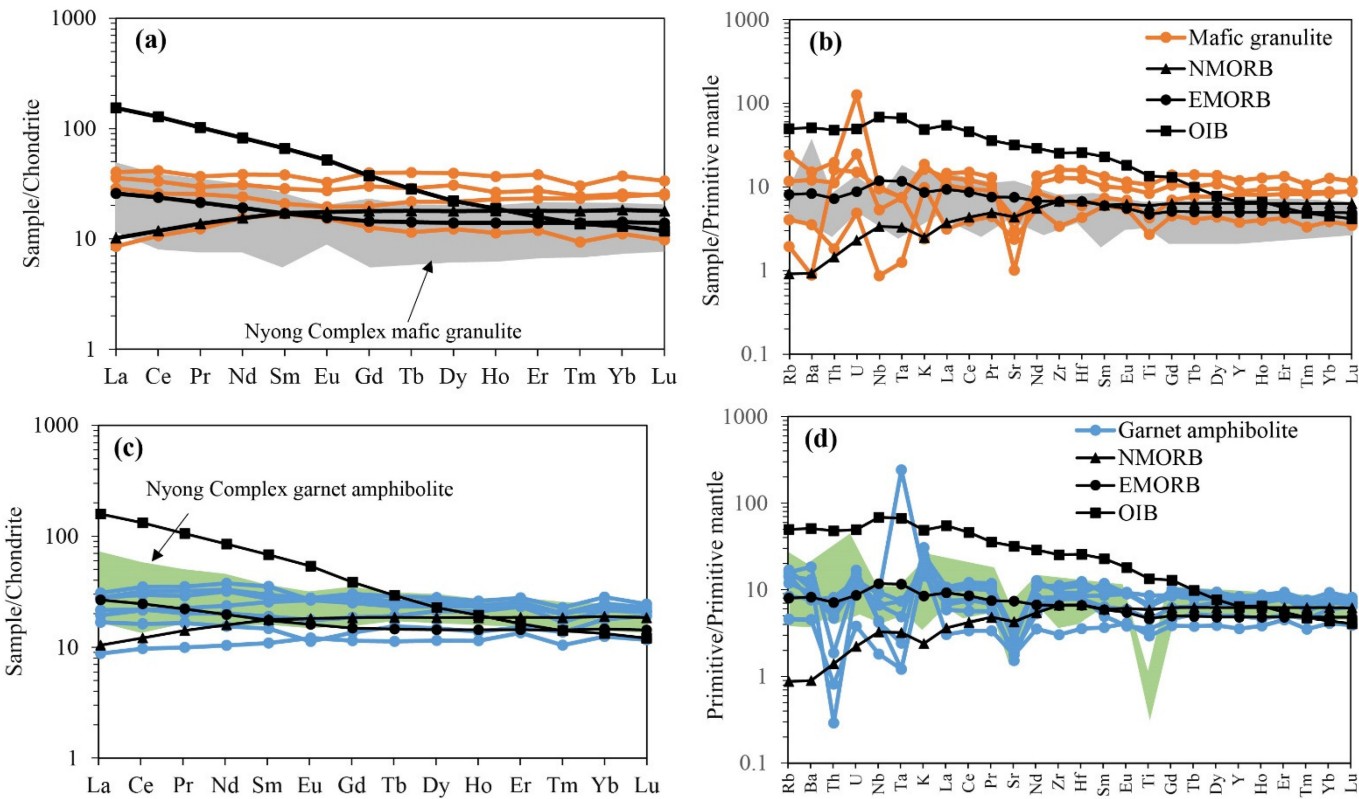

**Figure 8.** (**a**,**c**) Chondrite-normalized (normalization values after [55] REE plots for the Anyouzok mafic granulite and garnet amphibolite, respectively); (**b**,**d**) primitive mantle-normalized (normalization values after [52] multielement diagrams for the Anyouzok mafic granulite and garnet amphibolite garnet amphibolite, respectively). Nyong Complex mafic granulite data are from [11,31]. Nyong Complex garnet amphibolite data are from [22].

Trace and rare earth elements

The Anyouzok IFs generally show low trace element contents, <10 ppm. However, relatively high values are observed in some LILEs, such as Sr (6.6–11.5ppm; 5.0–20.7 ppm), Rb (23.6–88 ppm; 2.8–33.7 ppm), and Ba (39–74 ppm; 3.8–134 ppm) contents for BIFs and SBIFs, respectively. HFSEs such as Zr (12.4–18 ppm; 3.7–7 ppm) and Th (1.1–1.5 ppm; <0.05–0.16 ppm) present higher contents in BIFs compared to SBIFs, respectively.

The total REE-Y concentrations are higher for BIFs (37.43–39.69 ppm) compared to SBIFs (17.25–24.4 ppm). The chondrite-normalized [55] plot shows homogenous patterns with LREE enrichment over the HREE (Figure 9a) and positive Eu anomalies $((Eu/Eu^*)_{CN})$ ranging from 1.23 to 1.66 and from 1.52 to 1.98 for BIFs and SBIFs, respectively. PAAS-normalized REE-Y plots (normalization after [53]) are consistent for both BIFs and SBIFs (Figure 9b). Heavy rare earth elements (HREE) are enriched over light rare earth elements (LREE) with prominent positive Eu anomalies $(Eu/Eu^*)_{SN}$ ranging from 1.86 to 2.68 in BIFs and from 2.14 to 3.17 in SBIFs. Y/Ho ratios range from 26.4 to 30.33 for BIFs and from 27.50 to 38.33 for SBIFs. Pr/Yb ratios vary between 1.22 and 2 and between 0.65 and 2.24 for BIFs and SBIFs, respectively. BIFs show positive La $((La/La^*)_{SN} = 1.1–1.23)$ and Gd $((Gd/Gd^*)_{SN} = 1.21–1.35)$ anomalies, whereas the SBIFs display positive Gd $((Gd/Gd^*)_{SN} = 1.03–1.45)$ and positive to negative La $((La/La^*)_{SN} = 0.75–2.66)$ anomalies. Except for one SBIF sample (IS54), all the analyzed IFs present negative Y anomalies $((Y/Y^*)_{SN} = 0.80–0.89$ and $0.87–0.94)$.

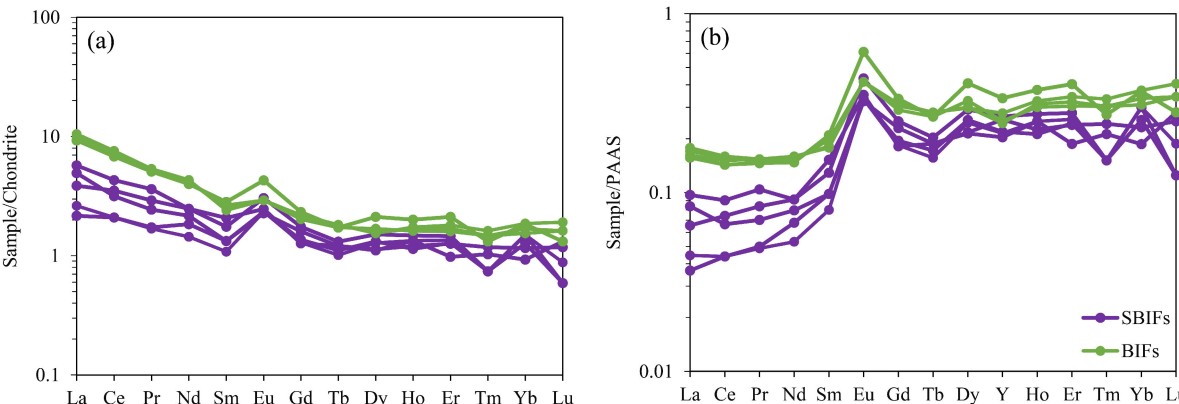

**Figure 9.** Chondrite—(**a**) and (**b**) PAAS-normalized REE-Y of the Anyouzok iron formations (BIFs and SBIFs). Normalization values after [53,55].

## 6. Discussion

### 6.1. Alteration, Metamorphism, and Element Mobility Assessment

Most Precambrian rocks have been affected by processes such as metamorphism, metasomatism, and deformation, which tend to modify their primary geochemical features [56–59]. In the current study, field investigations combined with petrographic and geochemical studies show that the rocks from the Anyouzok deposit have been subjected to some deformation, high-grade metamorphism, and alteration. Similar processes are widely reported within the whole Nyong Complex [11,12,20,24–26,28,30,31,60]. In this regard, the effects of post-emplacement processes on the mobility of major, trace, and rare earth elements should be assessed before any petrogenetic and geodynamic interpretations.

#### 6.1.1. Effects on Metavolcanic Rocks

The Anyouzok metavolcanic rocks have experienced high-grade metamorphism and some alteration, as reflected by the retrograde transformation of pyroxene into amphibole (Figure 5c) and the occurrence of secondary sericite and chlorite (Figure 5b,f). However, they mainly show low LOI (mean: 0.37 wt% and 0.90 wt% for mafic granulite and garnet amphibolite samples, respectively), indicating insignificant hydration or alteration during post-igneous processes, except for one mafic granulite sample (IS26 (LOI: −0.97 wt%)) and one garnet amphibolite sample (IS27b (LOI: −0.15 wt%)). The degrees of alteration of the analyzed metavolcanic rock samples were quantified using the chlorite-carbonate-pyrite index (CCPI; [61]) and the Ishikawa alteration index (AI; [62]). The analyzed rocks show relatively low AI (33.14–39.86 and 35.21–52.67) and moderate CCPI (50.53–67.51 and 52.88–71.00) for mafic granulite and garnet amphibolite, respectively, indicating minor to moderate alteration. In the CCPI vs AI diagram (Figure 10), the overall samples plot within the least altered box for mafic to felsic rocks, although some samples follow the chlorite-pyrite-(sericite) alteration trend, which also suggests some degrees of alteration and weak compositional modification of the major elements. Polat et al. [56] proposed that metavolcanic rock samples with 0.90 < Ce/Ce* < 1.10 lack LREE mobility, while samples with 0.90 > Ce/Ce* > 1.10 had undergone high LREE mobility. In the case of the Anyouzok metavolcanic rocks, Ce anomalies range from 0.96 to 1.08 and from 0.97 to 1.07, respectively, for mafic granulite and garnet amphibolite (Table 1), consistent with LREE immobility [56]. Furthermore, according to these previous authors, a positive correlation between Zr and other elements suggests the lack of mobility of these elements via alteration, since Zr is generally considered to be immobile. The analyzed samples show positive correlations with REEs (such as La, Ce, Sm, Eu, Gd, Dy, and Yb; not shown) and HFSEs (such as Nb, Y, and Hf; not shown) and scattered data points with some LILEs (such as Ba and Rb; not shown). In addition, their REE and HFSE patterns (Figure 8) are generally homogeneous and coherent, suggesting insignificant mobility during post-igneous metamorphism and

alteration. Therefore, immobile elements were considered to depict the igneous affinities, petrogenesis, and tectonic setting of the investigated metavolcanic rocks.

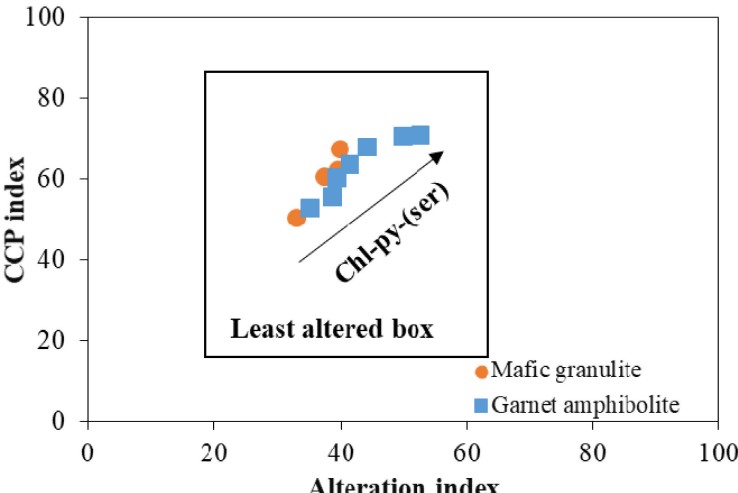

**Figure 10.** Alteration box plot [61] for the Anyouzok metavolcanic rocks. Overall samples fall within the least altered box and follow the chlorite alteration trend.

6.1.2. Effects on IFs

The effects of hydrothermal alteration on BIFs within the Nyong Complex have been recently reported by a few authors [27,29]. Polished thin section observations of the Anyouzok IFs revealed secondary minerals such as calcite, pyrite, hematite, and chlorite (Figure 6c,e). These secondary minerals are mainly encountered in SBIFs, which have experienced shearing, but are less obvious in BIFs. Geochemical data present a general increase in $Fe_2O_3$ and a decrease in $SiO_2$ within the SBIFs, in contrast to BIFs (except for sample IS13), resulting in the increase in Fe/Si ratios from 0.67 in BIFs to 1.72 in SBIFs. Moreover, the SBIFs matrix presents numerous cavities (Figure 6f), which are most likely a result of the leaching of silica [2]. Therefore, the ubiquitous shearing that affected the Anyouzok SBIFs may have created paths for fluid circulation, facilitating hydrothermal alteration and leaching processes. The formation of Ca-bearing minerals such as calcite or epidote are generally linked with hydrothermal alteration [10]. In this view, these authors proposed the use of $Fe_2O_3/Al_2O_3$ vs. $CaO/Al_2O_3$ and $Fe_2O_3/TiO_2$ vs. $CaO/TiO_2$ binary diagrams to assess the effect of hydrothermal alteration. In these diagrams (Figure 11), the SBIF samples show strong positive correlations ($r^2 = 0.99$), suggesting that the increase in $Fe_2O_3$ concentrations is linked with hydrothermal alteration. In contrast, such correlations are not observed in BIFs. In addition, PAAS-normalized REE-Y patterns of the investigated IFs (Figure 9b) exhibit prominent positive Eu anomalies and HREE enrichment over LREE, comparable to many Archean to Paleoproterozoic IFs worldwide [6,8,10,15,54,60,63], suggesting that most samples kept their primary REE-Y systematics. Based on the above discussion, we suggest that BIFs are more reliable in determining the characteristics of the Anyouzok IFs during their deposition, while SBIFs characteristics should be used with caution.

*6.2. Petrogenesis of the Metavolcanic Rocks*

Considering previous investigations on the geochemical features of metabasic rocks within the Nyong Complex greenstone belts [11,28,30,31,37,64], it is suggested in the current study that the Anyouzok metabasic rocks are of volcanic origin. In the Zr/Ti vs. Nb/Y plot [65], all the Anyouzok metavolcanic rock samples, along with the Kribi and Bipindi mafic granulites, show basaltic compositions (Figure 12a). In contrast, the Akom II garnet amphibolites show more evolved compositions and fall in the rhyodacitic rocks field. The Anyouzok metavolcanic rock samples mainly show flat patterns in chondrite-normalized REE diagrams (Figure 8a,c), suggesting that they are of the tholeiitic series. In the Zr vs. Y

diagram (Figure 12b; [66]), their data points show tholeiitic to transitional affinities, similar to the Kribi and Bipindi mafic granulites and the Akom II garnet amphibolites.

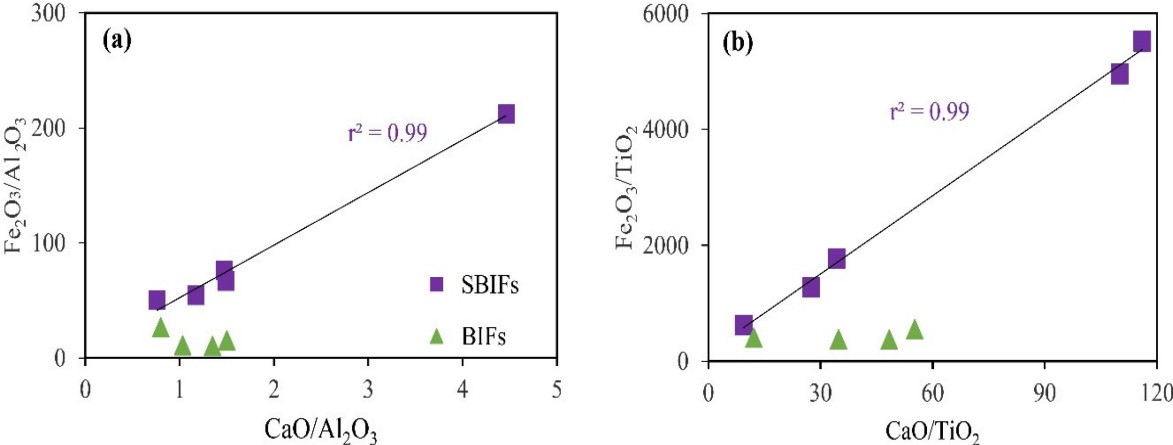

**Figure 11.** (**a**) $Fe_2O_3/Al_2O_3$ vs. $CaO/Al_2O_3$ and (**b**) $Fe_2O_3/TiO_2$ vs. $CaO/TiO_2$ binary plots [10]; hydrothermal alteration effect on the Anyouzok IFs.

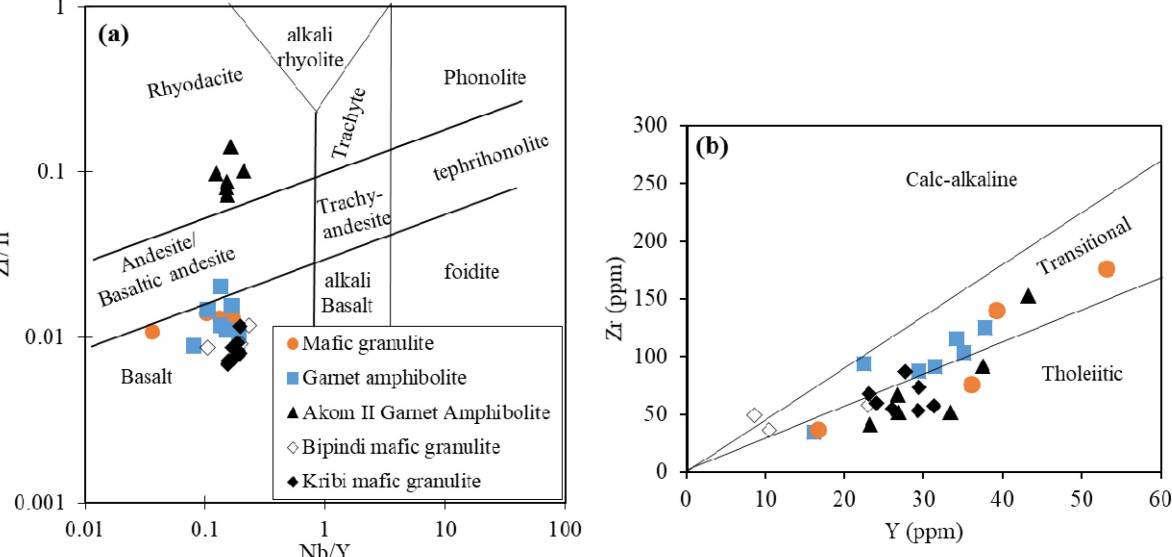

**Figure 12.** Classification plots for the Anyouzok metavolcanic rocks: (**a**) $Zr/TiO_2$ vs. Nb/Y [65]; (**b**) Y vs. Zr plot [66].

Mafic to ultramafic source magmas tend to assimilate crustal components during their ascent to the surface, resulting in variable degrees of crustal contamination [67,68]. The extent and nature of this input could be evaluated using elemental concentrations and various ratios, showing different variations in crustal- and mantle-derived materials [52,67–70]. For instance, the troughs in Nb and Ta exhibited by most analyzed samples in multi-element diagrams (Figure 8b,d) suggest crustal input [67–69]. The Th/Nb (0.19–0.36; mean: 0.24 and 0.004–0.16; mean: 0.08 for mafic granulite and garnet amphibolite, respectively) and Nb/Y (0.036–0.17; mean: 0.11 and 0.08–0.19; mean: 0.14 for mafic granulite and garnet amphibolite, respectively) ratios of the analyzed samples are lower than those of UCC (Th/Nb: 0.87; Nb/Y: 0.57; [70]), suggesting insignificant to minor crustal contamination.

The Anyouzok metavolcanic rocks present variable MgO (4.01−11.4 wt%) and Mg# (26.98−64.63) contents, suggesting that their precursor primary melts experienced fractional crystallization during magma ascent [71]. This is further supported by the variable Cr contents (20−820 ppm) of the analyzed samples, with values lower than those of mantle-derived melts (Cr >1000 ppm; [71]), suggesting some degree of fractional crystallization.

The MgO contents show a wide compositional range within the dataset, which serves as an index of differentiation in the binary plots of mafic rocks [71]. When plotted against MgO, Cr shows a strong positive correlation ($r^2 = 0.98$) and a weak positive correlation ($r^2 = 0.67$) for mafic granulite and garnet amphibolite, respectively (Figure 13a), suggesting a higher degree of clinopyroxene and/or spinel fractionation within mafic granulite samples. CaO decrease with decreasing MgO for the mafic granulite, indicating the fractionation of clinopyroxene (Figure 13b). Such fractionation lacks in garnet amphibolite, as suggested by their scattered data points (Figure 13b), which could be attributed to Ca mobility. Moreover, clinopyroxene fractionation in the mafic granulite samples is also depicted via the increase in $SiO_2$ with decreasing MgO (Figure 13c). The chondrite-normalized REE diagrams (Figure 8a,c) show slightly negative to null Eu anomalies (Eu/Eu*)$_{CN}$ = 0.84–1.04 and 0.80–1.09 for mafic granulite and garnet amphibolite, respectively), suggesting minor to no plagioclase fractionation. The troughs in Sr observed in multi-element diagrams (Figure 8b,d) could reflect post-magmatic processes such as alteration or metamorphism. Moreover, the relatively flat chondrite-normalized REE patterns (Figure 8a,c) presented by most samples also suggest minor fractional crystallization. Based on these characteristics, the investigated rocks were plotted in the La/Yb vs. Yb [72] binary diagram (Figure 13d) and indicated that fractional crystallization is subordinate to partial melting in the parental melt genesis.

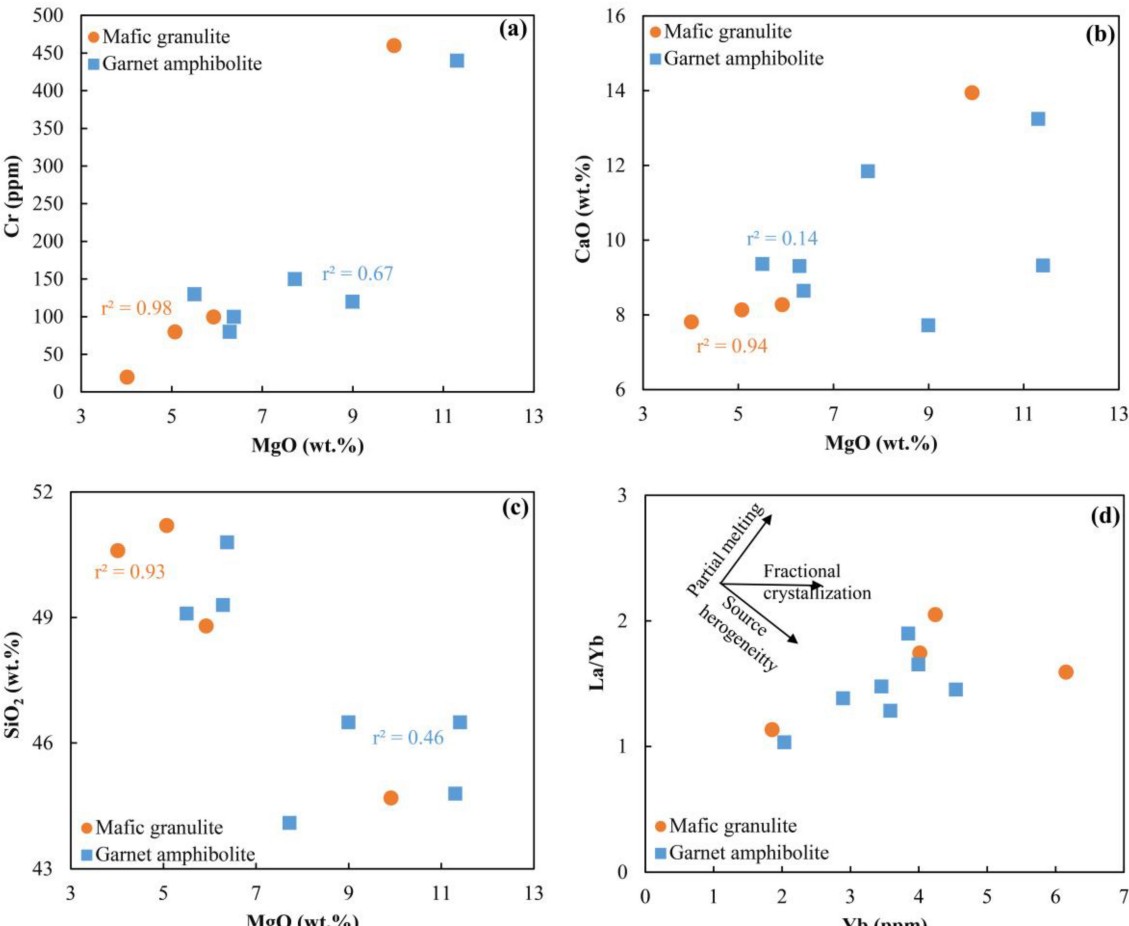

**Figure 13.** Binary plots of (**a**) Cr vs. MgO; (**b**) CaO vs. MgO; (**c**) $SiO_2$ vs. MgO; and (**d**) Yb vs. La/Yb [72].

Compositional variations of REE and HFSE and their elemental ratios generally assist in the depiction of the source and melting conditions of mantle melts [73–76]. For instance, the flat HREE patterns of most analyzed samples (Figure 8a,c), Dy/Yb (1.37–1.84 and 1.27–1.65) and the (Gd/Yd)$_{CN}$ (0.82–1.17 and 0.76–1.19) ratios of the mafic granulite and

garnet amphibolite samples, respectively, point toward a shallow mantle source lacking residual garnet [73,74]. Excluding one mafic granulite sample (IS20), the Zr/Nb (20.54–25.51 and 17.42–31.33) and Zr/Hf (36.67–42.22 and 30.91–37.60) ratios of mafic granulite and garnet amphibolite, respectively, are comparable to the primitive mantle (Zr/Nb = 15.71 and Zr/Hf = 36.25; [52]) and NMORB (Zr/Nb = 31.76 and Zr/Hf = 36.10; [52]), suggesting a depleted mantle source. Moreover, the chondrite-normalized REE diagrams of the Anyouzok metavolcanic rocks show MORB-like patterns (Figure 8a,c), also hinting at source derivation from a depleted mantle.

In the Nb vs. Zr binary diagram proposed by [75], the analyzed samples, similar to the Nyong Complex mafic granulite and garnet amphibolite samples, plot within or close to the depleted mantle source field (Figure 14a). Furthermore, in the Dy/Yb vs. La/Yb diagram (Figure 14b; [73]), the data points mainly indicate ca. 4% partial melting of a spinel-peridotite source, with no residual garnet. Primitive mantle-normalized multi-element diagrams of the Anyouzok metavolcanic rocks present HFSEs (Nb, Ta, Zr, and Ti) depletion for most samples, coupled with relatively high Th/Yb (0.08–0.39; 0.006–0.15) and La/Nb (1.32–3.5; 0.98–1.65) ratios compared to those of depleted mantle (Th/Yb = 0.02 and La/Nb = 1.29; [77]). These features generally reflect the source metasomatism of the precursors by melts or by subduction-related fluids [68,71]. In this view, the analyzed samples were plotted in the Rb/Y vs. Nb/Y [78] diagram (Figure 14c) and show that the mantle source of the Anyouzok metavolcanic rock precursors slightly experienced fluid metasomatism. Based on these geochemical characteristics, it is suggested that the primary melt of the Anyouzok metavolcanic rock protoliths originated from shallow depth partial melting of a slightly metasomatized spinel peridotite source, which has experienced various degrees of fractional crystallization and minor crustal contamination. A comparable origin has been proposed for precursors of metabasic to ultrabasic rocks also occurring within the Nyong Complex [24,30,31].

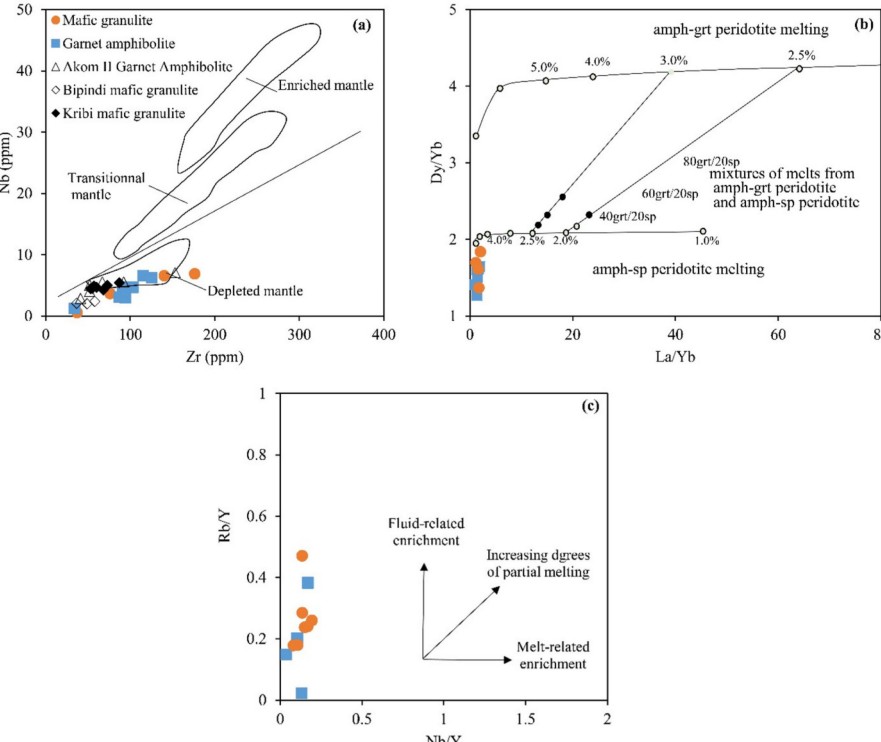

**Figure 14.** Binary plots for magma source characteristics of the Anyouzok metavolcanic rocks: (**a**) Zr vs. Nb [75]; (**b**) La/Yb vs. Dy/Yb [73]; (**c**) Rb/Y vs. Nb/Y [78].

### 6.3. Tectonic Setting of the Metavolcanic Rocks

Previous workers reported various tectonic environments for the Nyong Complex metabasic rocks, including back-arc, within-plate, NMORB, EMORB, and OIB settings [11,28,30,31,64]. Chondrite-normalized diagrams show that the Anyouzok metavolcanic rocks mainly show NMORB and EMORB affinities (Figure 8a,c). Furthermore, in the primitive mantle-normalized multi-element plots (Figure 8b,d), most samples exhibit depletion in Nb, Ta, Ti, and Zr, which are commonly observed in the arc basalt setting [79–81]. Various tectonic discrimination diagrams based on immobile elements are generally used to constrain the geodynamic setting of metamorphosed mafic rocks [68,82]. In the La/10-Nb/8-Y/15 ternary diagram (Figure 15a) proposed by [82], the studied garnet amphibolite samples, like the Akom II garnet amphibolite plotted for comparison, show back-arc and EMORB characteristics. In contrast, the analyzed mafic granulite samples mainly have arc tholeiites and back-arc features, similar to other Nyong Complex mafic granulites. In the Th/Yb vs. Nb/Yb discrimination diagram (Figure 15b), the Anyouzok samples are scattered within and above the mantle array. The garnet amphibolite samples fall within the back-arc field and NMORB area, while the mafic granulite samples plot along the arc field and follow the within-plate enrichment trend defined by [68]. Based on these geochemical features and earlier studies of metabasic rocks within the Nyong Complex, an association between back-arc and arc setting is suggested for the emplacement of the Anyouzok metavolcanic rocks precursors.

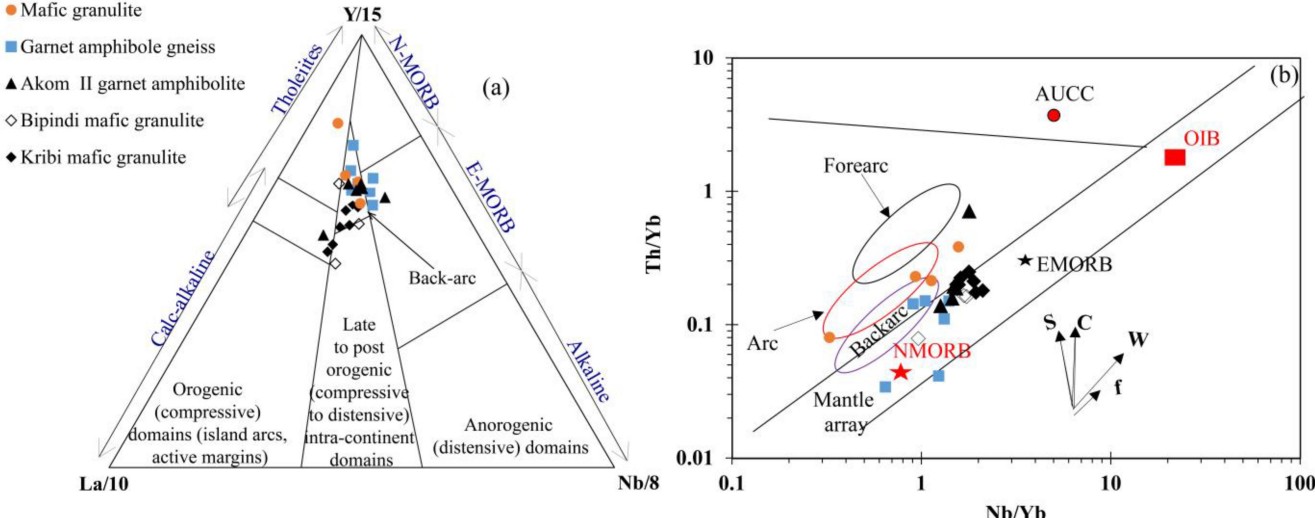

**Figure 15.** Tectonic discrimination plots for the Anyouzok metavolcanic rocks: (**a**) La/10-Nb/8-Y/15 [82]; (**b**) Th/Yb vs. Nb/Yb. NMORB: Normal Mid-Ocean Ridge Basalts, E-MORB: Enriched Mid-Ocean Ridge Basalts, OIB: Ocean Island Basalts, AUCC: Archean Upper Continental Crust. The vectors f, W, C, and S refer to fractional crystallization, within-plate fractionation, crustal contamination, and subduction zone.

### 6.4. Nature, Source, and Assessment of Detrital Input during the Anyouzok IF Deposition

The primary features of IFs could be influenced by the occurrence of clastic and/or volcanic components, yielding to high contents of some trace elements generally considered immobile and which lack in seawater (e.g., $Al_2O_3$, $TiO_2$, Zr, Th, Nb, and Sc; [6,8,9,83–85]. In addition, these detritus produce correlations between the former listed immobile elements and some REE and HFSE ratios such as Y/Ho and Pr/Yb [8]. A petrography study has revealed the presence of detrital components within the Anyouzok IFs depicted by minerals such as plagioclase and K-feldspar (Figure 6b,c,e,f). The Anyouzok BIFs show $Al_2O_3$ (mean: 2.14 wt%), $TiO_2$ (mean: 0.08 wt%), and Zr (mean: 16.08 ppm) contents and Pr/Yb ratios (mean: 1.76), suggesting a slight detritus contribution during the deposition of BIFs.

The estimation of the detrital input in the Anyouzok IFs was based on the Fe/Ti vs. Al/(Al + Fe + Mn) diagram by [86], displaying the ideal mixing between terrigenous and metalliferous sediments. In this plot (Figure 16a), the analyzed BIFs reveal up to 15% detritus in their composition, whereas the SBIFs reflect insignificant contamination for most samples (except for sample IS41, with ca. 10% crustal input). As discussed in Section 6.1.2., an increase in Fe within the SBIFs is associated with hydrothermal alteration, which facilitated leaching processes and, thus, the position of their data points being closer to the hydrothermal sediments area (Figure 16a). Th/U ratios < 5 have been proposed as an indicator of the presence of phosphate (e.g., apatite, monazite) and contaminants during chemical sediment deposition [8], since volcanic and clastic materials generally have values ranging from ca. 3–5 [52,70,87]. In the current study, phosphate minerals such as monazite or apatite have neither been identified in thin sections nor via XRD investigations. The studied BIFs have Th/U ratios ranging from 3.84–5.50, similar to those of volcanic and clastic materials [52,70,87]. In addition, Figure 16b indicates increasing Th/U ratios with decreasing Zr concentrations in the Anyouzok BIFs, suggesting the influence of volcanic or clastic detritus in the Anyouzok BIFs [8]. Lower concentrations and ratios, and a lack of correlations between Zr and Th/U presented by the SBIF samples (Table 2), could be attributed to the leaching of materials during hydrothermal activities [2]. To further evaluate the influence of detritus on the REE-Y systematics of the Anyouzok IFs, binary plots such as Zr and Pr/Yb vs. Y/Ho (La/La*)$_{SN}$ and (Ce/Ce*)$_{SN}$ were used (Figure 16c–h). In Figure 16e, (Ce/Ce*)$_{SN}$ highlighted the positive correlation against Zr for the SBIFs and no correlation against Pr/Yb. In contrast, no other important positive correlation is observed with Y/Ho and (La/La*)$_{SN}$, suggesting that the presence of detritus and/or alterations have not significantly influenced the REE-Y systematics of the Anyouzok IFs.

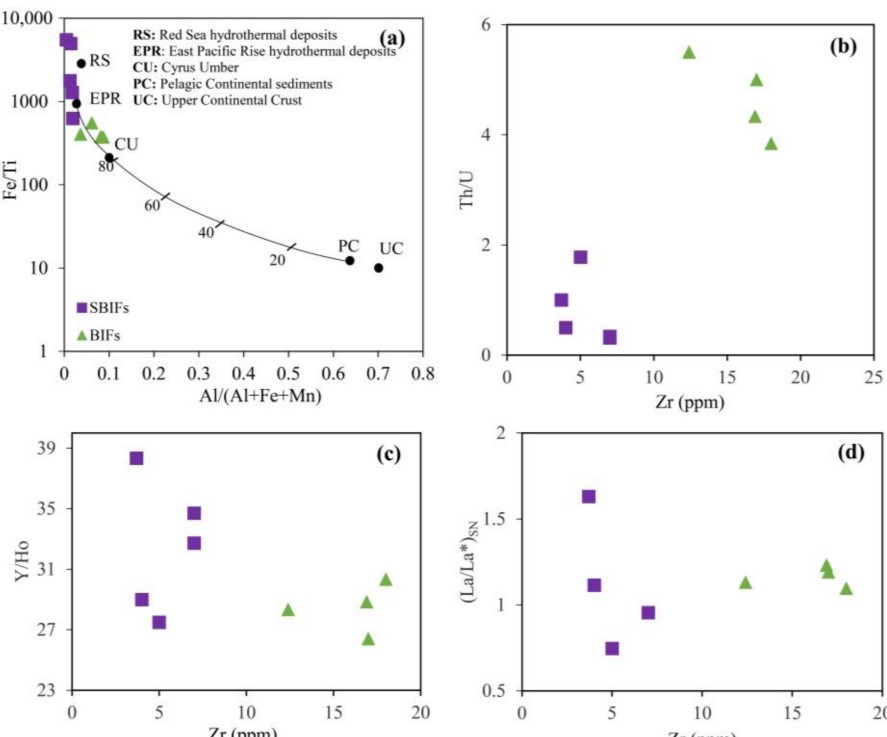

**Figure 16.** *Cont.*

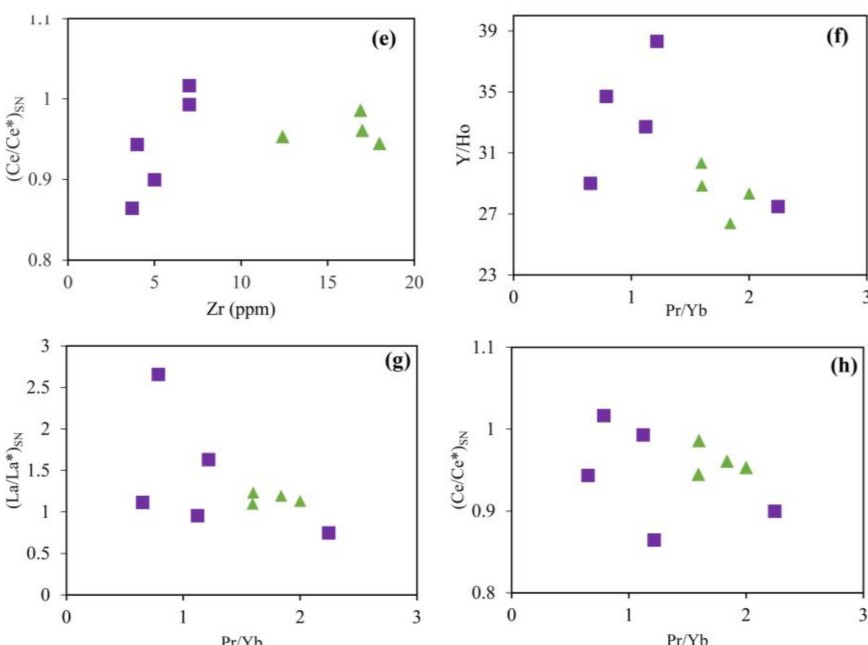

**Figure 16.** Detrital input assessment of the Anyouzok IFs: (**a**) Fe/Ti vs. Al/(Al + Fe + Mn). The curve represents the mixing of pelagic sediments (PC) with East Pacific Rise deposits (EPR); the approximate amount of EPR in the mixture [88] is indicated by the numbers in percentages. CU: Cyprus umber, UC: Upper continental crust, RS: Red sea hydrothermal deposits [53,89]; (**b**) Th/U vs. Zr; (**c**) Y/Ho vs. Zr; (**d**) (La/La*)$_{SN}$ vs. Zr; (**e**) (Ce/Ce*)$_{SN}$ vs. Zr; (**f**) Y/Ho vs. Pr/Yb; (**g**) (La/La*)$_{SN}$ vs. Pr/Yb; (**h**) (Ce/Ce*)$_{SN}$ vs. Pr/Yb.

## 6.5. Origin of the Anyouzok Iron Formations

### 6.5.1. Seawater and Hydrothermal Fluid Contribution

REE-Y systematics is widely used to assess the origin of Si and Fe in IFs [6,7,15,54,83–85,90]. Chemical sediments originating from seawater are characterized by super-chondritic Y/Ho ratios (>44), shale-normalized REE-Y patterns exhibiting positive La, Y, and Gd anomalies, negative Ce anomalies, and HREE enrichment over LREE and MREE [6,8,54]. In PAAS-normalized REE-Y diagrams, the Anyouzok IFs show consistent patterns (Figure 9b), with LREE depletion over HREE, positive Gd and La anomalies (except for two SBIF samples: IS19 and IS41), and negative Y anomalies (except for one SBIF sample: IS54). They show chondritic to super-chondritic Y/Ho ratios in both BIF (26.40–30.33) and SBIF (27.5–38.33) samples. Almost all BIF samples (except for IS13) exhibit negative Y anomalies and chondritic Y/Ho ratios, indicating they could be derived from slow rates of Fe oxyhydroxide precipitation [83,91,92].

Several workers reported that prominent positive Eu anomalies in IFs reflect the influence of high-temperature hydrothermal fluids, whereas a lack of Eu anomalies is considered reflective of low-temperature hydrothermal fluids [6,84,85,92–97]. The Anyouzok IFs show positive (Eu/Eu*)$_{SN}$ anomalies ranging from 1.86–2.68 and from 2.14–3.17 for the BIFs and SBIFs, respectively, suggesting the influence of high-temperature hydrothermal fluids. In addition, [54] reported that high-temperature hydrothermal fluids (>250 °C) have (Eu/Eu*)$_{CN}$ > 1, while low-temperature hydrothermal fluids (<250 °C) have (Eu/Eu*)$_{CN}$ ≈ 1. In this view, the Eu anomalies ((Eu/Eu*)$_{CN}$) of 1.23–1.66 and 1.52–1.98 observed for the analyzed BIFs and SBIFs, respectively, could account for high-temperature hydrothermal fluids contribution. However, in the (Eu/Eu*)$_{SN}$ vs. LREE diagram (Figure 17a), excluding sample IS13, showing secondary enrichment, all BIFs samples lack correlation, suggesting hydrothermal fluids input during their precipitation [6,84,85,92,93]. In contrast, the increase in Eu anomalies with decreasing LREE in SBIFs samples could indicate the influence of post-depositional processes, such as hydrothermal alteration, as previously discussed in Section 6.1.2. [84,85] proposed binary diagrams based on the Eu/Sm, Y/Ho, and Sm/Yb

ratios and defined the mixing line using seawater and high-temperature hydrothermal fluid end-members in order to evaluate their contribution in the solute sources of IFs. Figure 17b shows that small quantities of high-temperature hydrothermal fluids (0.1%) could account for the Eu/Sm and Y/Ho ratios presented by the Anyouzok IFs. Furthermore, the analyzed IFs fall along the mixing line (Figure 17c) defined by [85] and suggest an input of ca. 5% of high-temperature hydrothermal fluids in the Anyouzok BIFs and that they were deposited distal to the hydrothermal vent. It is therefore suggested that a mixture of seawater, low proportions of high-temperature hydrothermal fluids, and variable quantities of detrital materials influenced the deposition of the Anyouzok IFs. Comparable results were reported for several IF occurrences within the Congo Craton in Cameroon [12,25,27,63,98] and Congo [13,15].

### 6.5.2. Palaeoredox State of IF

Ce anomalies in chemical sediments are widely used to assess the palaeoredox state of the ancient seawater [7,17,54,99]. Suboxic and anoxic seawaters lack negative Ce anomalies, unlike oxygenated seawater, which displays strong negative Ce anomalies [7,54]. To distinguish "true" from "false" negative Ce anomalies, the $(Ce/Ce^*)_{SN}$ vs. $(Pr/Pr^*)_{SN}$ diagram (Figure 17d) has been proposed by [54]. In this plot, most of the Anyouzok IF samples show no Ce anomalies, suggesting that they were deposited in a suboxic to anoxic environment. Only the SBIF sample IS41, with a negative La (0.75) anomaly, plots within the oxic environment field. This could be attributed to post-depositional processes, which affected the SBIFs, as discussed in Section 6.1.2, since IFs deposited in the oxic environment show positive La anomalies (e.g., [6,96,97]). Therefore, we suggest that the Anyouzok IFs were deposited under anoxic conditions, similar to most of the Nyong Complex and Archean to Paleoproterozoic IFs worldwide [7,12,13,47,63,90,100].

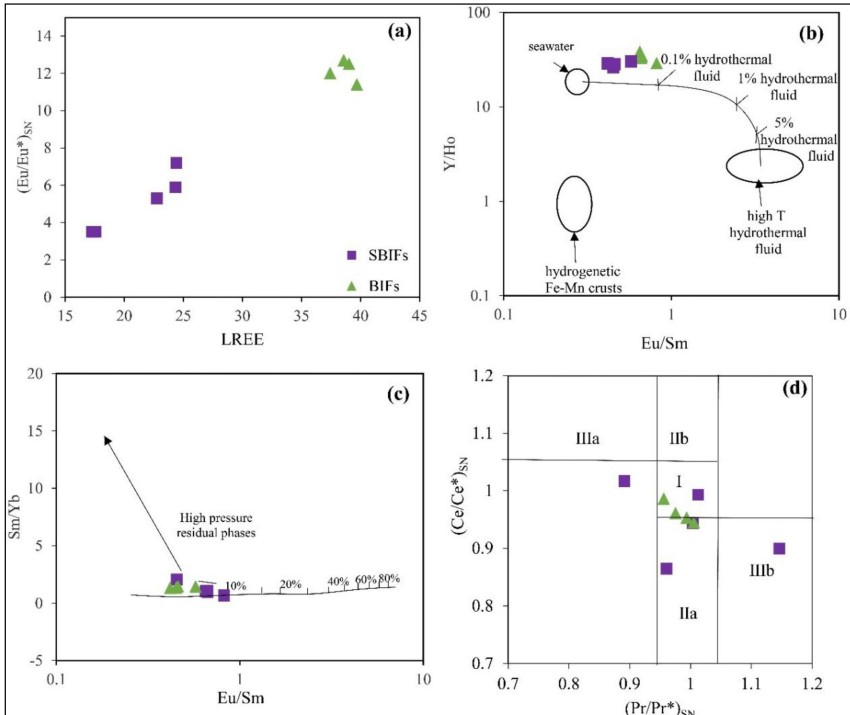

**Figure 17.** Contribution of seawater and hydrothermal fluids for the Anyouzok IFs precipitation: (**a**) $(Eu/Eu^*)_{SN}$ vs. LREE diagram for the Anyouzok IFs; (**b**) Eu/Sm vs. Y/Ho [84] with a conservative mixing line of high-temperature hydrothermal fluid [92] and seawater [101]; (**c**) Eu/Sm vs. Sm/Yb plot [85] with a conservative mixing line of high-temperature hydrothermal fluid [8] and seawater [101]; (**d**) $(Ce/Ce^*)_{SN}$ vs. $(Pr/Pr^*)_{SN}$ plot [54] for the Anyouzok IFs. I: neither Ce nor La anomaly; IIa: positive La anomaly, no Ce anomaly; IIb: negative La anomaly, no Ce anomaly; IIIa: positive Ce anomaly; IIIb: negative Ce anomaly.

*6.6. Depositional Setting*

Based on associated or interbedded rocks, IFs have been classified as Algoma-type when deposited close to volcanic centers within greenstone belts or as Superior-type when deposited distal from volcanic centers, on continental shelves, or on submerged platforms [17,18,102]. Previous workers suggested both superior- [25,27,60] and Algoma-type [11] affinities for the Nyong Complex IFs. Furthermore, several depositional environments were proposed for Nyong Complex IFs, including the island arc setting with MORB-like signatures [28], the volcanic arc setting [12], back-arc or continental margin sea environments [25,60], a large basin between a continental margin and an oceanic volcanic center [27], and an extensional basin between a continental margin and a back-arc setting [11]. The Anyouzok IFs are interbedded with metavolcanic rocks comprising mafic granulite and garnet amphibolite along the stratigraphy (Figure 4a and b), suggesting an Algoma-type deposit. Considering the conspicuous higher Shale-normalized Eu anomalies of the Algoma type IFs, [103] proposed a threshold with $(Eu/Eu^*)_{NASC} > 1.8$ for Algoma-type IFs and values <1.8 for the superior-type IFs. On this view, the high $(Eu/Eu^*)_{NASC}$ anomalies presented by the studied BIFs (1.88–2.7) and SBIFs (2.16–3.26) suggest an Algoma-type deposit for the Anyouzok IFs, although the higher values presented by SBIFs may reflect the influence of hydrothermal alteration. Taking into consideration our results and those of previous IF investigations within the Nyong Complex, we suggest that the Anyouzok IFs are Algoma-type and were deposited distal to the hydrothermal vents in the back-arc setting.

## 7. Conclusions

This work integrates field and petrographic investigations, coupled with whole rock geochemical data of rocks from the Anyouzok area in the Nyong Complex greenstone belts. The following conclusions can be provided:

1.  The lithostratigraphy of this area comprises an IF unit, consisting of BIFs and SBIFs, and a country rock unit made up of mafic granulite and garnet amphibolite. These rocks were intensely deformed and metamorphosed up to granulite facies. BIFs are absent in surface outcrops and were uniquely intercepted in one drillhole, sandwiched between mafic granulites.
2.  The Anyouzok metavolcanic rocks have tholeiitic to transitional basalt precursors. The latter originated from the partial melting of a metasomatized spinel lherzolite source, which experienced various degrees of fractional crystallization and was emplaced in an arc/back-arc setting.
3.  The Anyouzok IFs are mainly composed of magnetite, quartz, and metamorphic amphibole (actinolite and tremolite), with subordinate biotite, plagioclase, K-feldspar, hematite, pyrite, calcite, and ilmenite. These rocks were primarily deposited as BIFs and enriched to SBIFs through hydrothermal alteration activities and the leaching of silica.
4.  The Anyouzok BIFs recorded the contribution of detrital components during their deposition. In addition, the REE-Y systematics of both BIFs and SBIFs suggest the influence of seawater and high-temperature hydrothermal fluids distal to the vent source during their precipitation in an anoxic to suboxic environment. The ubiquitous negative Y anomalies and chondritic Y/Ho ratios observed in almost all BIF samples suggest slow rates of Fe oxyhydroxide precipitation.
5.  Based on the geochemical features of the Anyouzok IFs and interbedded metavolcanic rocks, we propose that these IFs are Algoma-type and were formed distal to the hydrothermal vents in a back-arc setting.

**Supplementary Materials:** The following supporting information can be downloaded at: https://www.mdpi.com/article/10.3390/min12101198/s1, Table S1: Anyouzok iron ore deposit drillhole details; Table S2: Anyouzok iron ore deposit logging details.

**Author Contributions:** Conceptualization, I.S.F., M.N.T. and D.H.F.; formal analysis, J.P.N., C.E.S., S.G. and I.S.F.; investigation, I.S.F. and J.P.N.; writing—original draft preparation, I.S.F., M.N.T., D.H.F. and L.S.T.; writing—review and editing, I.S.F., M.N.T., D.H.F., C.M., L.S.T., S.G., C.E.S. and J.P.N.; project administration, J.P.N.; funding acquisition, I.S.F. All authors have read and agreed to the published version of the manuscript.

**Funding:** This research did not receive any specific grant from funding agencies in the public, commercial, or not-for-profit sectors.

**Data Availability Statement:** The data used to support the findings of this study are available with the corresponding author upon request.

**Acknowledgments:** This paper represents part of the first author's PhD thesis at the Department of Earth Sciences of the University of Yaoundé I. The authors are grateful to the International Mining and Infrastructure Corporation PLC (IMIC) via Caminex for providing logistic support during the field work. We would also like to thank two anonymous referees for their constructive review comments that greatly improved the manuscript.

**Conflicts of Interest:** The authors declare no conflict of interest.

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
