# Peer review of "Lithostratigraphy, Origin, and Geodynamic Setting of Iron Formations and Host Rocks of the Anyouzok Region, Congo Craton, Southwestern Cameroon"

_minerals, doi:10.3390/min12101198_

Round 1

Reviewer 1 Report

This contribution is significant to understand the IFs mineralization of the Congo Craton. It also can provide the global information on the geochemistry of BIFs. The content is well organized. It can be published after a minor revision. Some problems are listed below:

1. Lines 46-47, “96.9 Mt @ 34.92% Fe”, what is @? Means “and”?

2. Fig. 1, Check the spelling of ages.

3. Fig. 5c, where is amphibole?

4. Fig. 5f, where is chlorite?

5. Line 279,shared” banded iron formations?

6. Fig. 6b-c, are you sure it can be identified as “Banded iron formation”?

7. Fig. 16a, what is ordinate value?

8. Line 714, IS41 might be an abnormal sample, as La should be positive anomalies in oxic environment (e.g., Bolhar et al., 2004; Frei et al., 2008, Tang et al., 2013).

9. Give a mineralization modal sketch for the Anyouzok IFs.

10. Part of comments and text corrections can be seen in the annotated MS (pdf-format).

Author Response

We appreciate the reviewer’s suggestions and constructive comments on the primary version of the manuscript. We have considered all the suggestions and comments of the reviewers for improving the manuscript. All the revisions are highlighted for your kind checking.

Some problems are listed below:

  1. Lines 46-47, “96.9 Mt @ 34.92% Fe”, what is @? Means “and”?

Reply: Thank you for the observation. “@” has been replaced with “at”

Fig. 1, Check the spelling of ages.

Reply: Thank you, the spelling of ages has been corrected.

  1. 5c, where is amphibole?

Reply: Thank you for highlighting. Fig. 5c has been replaced by a more explicit photomicrograph (Please see the revised Fig. 5c).

  1. 5f, where is chlorite?

Reply:Thank you for highlighting. Chlorite (Chl) has been indicated with arrows in the revised Figure 5f.

  1. Line 297, “shared” banded iron formations?

Reply: Thank you very much. “shared” has been replaced with “sheared.

  1. 6b-c, are you sure it can be identified as “Banded iron formation”?

Reply: Thank you for the observation. Fig. 6b has been replaced with a more appropriate photomicrograph showing the alternating quartz-rich and magnetite-rich bands of the Anyouzok Banded Iron Formation.

Yes, Fig. 6c is a section of the Anyouzok Banded Iron formation. It has been used to highlight mineral transformations observed within the rock.

  1. 16a, what is ordinate value?

Reply: Sorry for this mistake. Fig. 16a has been corrected accordingly.

  1. Line 448, e.g., Sun et al. (2014a,b).

Sun XH, Zhu XQ, Tang HS, Zhang Q, Luo TY. 2014a. The Gongchangling BIFs from the Anshan-Benxi  area, NE China: Petrological-geochemical characteristics and genesis of high-grade ores. Ore Geology Reviews. 60:112-125.

Sun XH, Zhu XQ, Tang HS, Zhang Q, Luo TY, Han T. 2014b. Protolith reconstruction and geochemical study on the wall rocks of Anshan BIFs, Northeast China: Implications for the provenance and tectonic setting. Journal of Geochemical Exploration. 136: 65-75.

Reply: Thank you very much, the references have been added in the revised manuscript.

  1. Line 682, See Chen and Fu (1991), Bolhar et al. (2004), Frei et al. (2008), Tang et al. (2013), Deng et al. (2014).

Bolhar R, Kamber BS, Moorbath S, Fedo CM, Whitehouse MJ. 2004. Characterisation of early Archaean chemical sediments by trace element signatures. Earth Planet. Sci. Lett. 222, 4360.

Chen YJ, Fu SG. 1991. Variation of REE patterns in early Precambrian sediments: theoretical study and evidence from the southern margin of the northern China Craton. Chin. Sci. Bull. 36, 11001104.

Deng XH, Chen YJ, Yao JM, Bagas L, Tang HS. 2014. Fluorite REE-Y (REY) geochemistry of the ca. 850 Ma Tumenmolybdenite-fluorite deposit, eastern Qinling, China: Constraints on ore genesis. Ore Geology Reviews. 63:532543.

Frei R, Dahl PS, Duke, EF, Frei KM, Hansen TR, Frandsson MM, Jensen LA. 2008. Trace element and isotopic characterization of Neoarchean and Paleoproterozoic iron formations in the Black Hills (South Dakota, USA): assessment of chemical change during 2.91.9 Ga deposition bracketing the 2.42.2 Ga first rise of atmospheric oxygen. Precambrian Res. 162, 441474.

Tang HS, Chen YJ, Santosh M, Zhong H, Yang T. 2013. REE geochemistry of carbonates from the Guanmenshan Formation, Liaohe Group, NE Sino-Korean Craton: Implications for seawater compositional change during the Great Oxidation Event. Precambrian Research, 227: 316336.

Reply: Thank you very much, the references have been added in the revised manuscript.

  1. Line 714, IS41 might be an abnormal sample, as La should be positive anomalies in oxic environment (e.g., Bolhar et al., 2004; Frei et al., 2008, Tang et al., 2013).

Reply: Many thanks for this comment which enables us to improve on the statement about the negative La anomaly of SBIF sample IS41.

  1. Give a mineralization modal sketch for the Anyouzok IFs.

Reply: We are grateful for the reviewer’s suggestion. Unfortunately, we did not collect IF ore samples for the Anyouzok iron deposit, which do not allow us to provide a comprehensive mineralization model sketch for the Anyouzok IFs. This will be conducted for future studies.

  1. Part of comments and text corrections can be seen in the annotated MS (pdf-format).

All the corrections have been properly applied as recommended by the referee.

Author Response

Geological setting

Line 112: typo in “Paleoproterozoic”

Reply: Thank you for picking out the typo. “Palaeoprotorozoic” has been replaced with “Paleoproterozoic”.

Methodology

There are major issues in this section:

Lines 234-249, it is said that “A total of 20 samples (5 SBIFs, 4 BIFs, 4 mafic granulite and 7garnet amphibolite) from eight drillholes were collected for this study”, but “twenty-one representative fresh samples (seven mafic granulite, five garnet amphibolite, six SBIFs and three BIFs) were selected for whole rock geochemical analysis”.

How it is possible?

Reply: We are thankful for your question that helps us to revise the manuscript properly. We revised and added more explanations to the methodology section (Please see revised section 4 ‘Sampling and analytical methods’).

According to the Fig. 4, the following samples have been investigated: mafic granulite: 7; garnet amphibolite: 5; SBIFs: 6; BIF: 1, totalizing 19 samples.

Reply: Thank you for this observation. In figure 4, there are actually 31 samples (8 SBIFs, 4 BIFs, 8 mafic granulites, and 11 garnet amphibolites) from eight drill holes (TH31, TD46, TD60, TE6, TE9, TD47, TE22, and TD65). Many of the sample points are not distinguishable in figure 4 due to the log scale.

According to the Table 1, there are: mafic granulite: 4; garnet amphibolite: 7; BIF: 4; SBIF: 5, totalizing 20 samples.

In other figures, the number of SBIFs plotted is variable. For instance, there are 5 SBIFs samples plotted in Fig. 16a, but only 4 in Fig. 16b.

Reply: There are 5 SBIFs and 4 BIFs samples investigated in the geochemical study. In Fig. 16b, two of the SBIFs samples are overlapping.

So, how many samples have been investigated? 19, 20 or 21? And what are the investigated samples?

Such confusion discredits the whole study.

Reply: Thank you very much for your keen observation. 20 samples (4 mafic granulites, 7 garnet amphibolites, 5 SBIFs, and 4 BIFs) were investigated (Please see revised section 4 ‘Sampling and analytical methods’).

More details should also be provided about the logging (how it has been done? What were

the rock feature surveyed, and how?)

Reply: Thank you for this comment. The sampling and analytical methods section has been improved in the revised manuscript (Please see revised section 4 ‘Sampling and analytical methods’).

The calculation of how Ce* and Eu* (and other anomalies) should be explained here, as well

as the normalizing values used for calculation.

Reply: Many thanks for your remarks. The formulas employed in the manuscript have been added to the sampling and analytical methods section of the revised manuscript (Please see revised section 4 ‘Sampling and analytical methods’).

Detail remarks:

Line 250: pulp?

Reply: Thank you for your keen observation. We have specified that it is rock pulp that was analyzed.

Line 255: at the beginning of a sentence, a number should better be written with letters.

Reply: Thank you. Modifications have been done in the revised manuscript

What are “ionic leach techniques”?

Reply: Many thanks for this comment. We apologize for this mistake. The analytical methods have been corrected in the revised manuscript (Please see revised section 4 ‘Sampling and analytical methods’).

Results

The positive Y anomalies, chondritic Y/Ho ratios and many other geochemical features indicate that the geochemical compositions of IFs (including REY) demonstrate a strong metamorphic overprint that completely modified initial IFs compositions. The high Al, Ti, Zretc., contents in iron formations are indicative of a rather important detrital input, preventing the use of trace elements to constrain the mode of formation of these IFs, or to make inference of coeval seawater chemistry. The authors themselves acknowledge that the chemical composition of most of their samples underwent a strong metamorphic overprint (their section 6.1, and specifically lines 510-511).

Reply: We are grateful for the reviewer’s comments. As discussed in section 6.1.2., the major elements composition and PAAS-normalized diagrams of the investigated IFs samples show that the Anyouzok BIFs have experienced minor modifications after metamorphism. In contrast, the SBIFs samples are use with caution, since they have strongly experienced hydrothermal alteration and leaching.

Except for one SBIF sample (IS54), both BIFs and SBIFs investigated in this study do not show positive Y anomalies, but slight negative Y anomalies.

For most of the BIFs samples (excluding IS13), the negative Y anomalies coupled with chondritic Y/Ho ratios indicating they could be derived from slow rates of Fe oxyhydroxide precipitation, as discussed by Bau (1999), Bau and Dulski (1999) and Basta et al. (2011).

You are right. The Anyouzok IFs show detrital component. However as discussed in section 6.4 and figure 16c-h, the lack of positive correlations between the selected elements indicate minor influence of detritus on the REE-Y systematics the Anyouzok BIFs.

All the above issues have been clarified in the revised manuscript

Discussion

The high metamorphic grade (amphibolite to granulite facies) and metamorphic texture displayed by the investigated samples indicate elemental mobility during metamorphic recrystallization. Thus, the discussion presented in the sections 6.2 and 6.3 should be presented much more cautiously. It seems that the authors recognize that their samples are altered, but they decided to conduct their interpretations as if the samples were unmetamorphosed and unaltered. This makes their interpretations and discussion questionable.

Despite this small discussion on post-depositional changes of the chemical composition of IFs, and despite the strong detrital component in IFs (section 6.4) the authors based their interpretations (section 6.5) on altered geochemical data.

The interpretations provided in the section 6.5 are thus fully unreliable

Reply: Thank you for this comment. As discussed in section 6.1 of our manuscript, prior to Petrogenetic interpretations, the investigated samples were subjected to alteration, metamorphism and element mobility assessment, which indicated insignificant mobility during post-igneous metamorphism and alteration. Therefore, immobile elements were considered to depict the igneous affinities, petrogenesis, and tectonic setting of the investigated metavolcanic rocks.

With regards to the Anyouzok iron formations (IFs), major elements composition and PAAS-normalized diagrams of the investigated IFs samples show that the Anyouzok BIFs have experienced minor modifications after metamorphism. In contrast, the SBIFs samples are use with caution, since they have strongly experienced hydrothermal alteration and leaching. Therefore, in our manuscript, we considered that BIFs are more reliable in determining the characteristics of the Anyouzok IFs during their deposition, while SBIFs characteristics were used with caution.

Overall, we found our results (both IFs and interbedded metavolcanic rocks) comparable with those reported in previous studies within the Congo craton (e.g. Maurizot et al., 1986; Lerouge et al., 2006; Ganno et al., 2016; Aye et al., 2017; Houketchang Bouyo et al., 2019; Ndema Mbongue et al., 2014; Ganno et al., 2015, 2016, 2017; Aye et al., 2017; Teutsong et al., 2017; Soh Tamehe et al., 2018, 2019, 2021, 2022; Ndime et al., 2019; Nga EssombaTsoungui et al., 2020; Nzepang Tankwa et al., 2021; Gatsé Ebotehouna et al., 2021; Gourcerol et al., 2022; Fuanya et al., 2019; Moudioh et al., 2020; Kwamou Wanang et al., 2021; Djoukouo Soh et al., 2021; Mvodo et al., 2022; Owona et al., 2022).

All the above issues have been clarified in the revised manuscript. The complete references of these works are as follow:

Aye, B.A., Sababa, E., Ndjigui, P.-D., 2017. Geochemistry of S, Cu, Ni, Cr and Au-PGE in the garnet amphibolites from the Akom II area in the Archaean Congo Craton, Southern Cameroon. Geochemistry 77, 81–93.

Djoukouo Soh, A., Ganno, S., Zhang, L., Soh Tamehe, L., Wang, C., Peng, Z., Tong, X., Nzenti, J.P., 2021. Origin, tectonic environment and age of the Bibole banded iron formations, northwestern Congo Craton, Cameroon: geochemical and geochronological constraints. Geol. Mag. 158, 2245–2263. https://doi.org/10.1017/S0016756821000765

Ganno, S., Moudioh, C., NzinaNchare, A., Kouankap Nono, G.D., Nzenti, J.P., 2016. Geochemical fingerprint and iron ore potential of the siliceous itabirite from PalaeoproterozoicNyong Series, Zambi area, Southwestern Cameroon. Resour. Geol. 66, 71–80.

Ganno, S., Ngnotue, T., KouankapNono, G.D., Nzenti, J.P., Notsa Fokeng, M., 2015. Petrology and geochemistry of the banded iron-formations from Ntem complex greenstones belt, Elom area, Southern Cameroon: Implications for the origin and depositional environment. Geochemistry 75, 375–387. https://doi.org/10.1016/j.chemer.2015.08.001

Ganno, S., Njiosseu Tanko, E.L., Kouankap Nono, G.D., Djoukouo Soh, A., Moudioh, C., Ngnotué, T., Nzenti, J.P., 2017. A mixed seawater and hydrothermal origin of superior-type banded iron formation (BIF)-hosted Kouambo iron deposit, Palaeoproterozoic Nyong series, Southwestern Cameroon: Constraints from petrography and geochemistry. Ore Geol. Rev. 80, 860–875. https://doi.org/10.1016/j.oregeorev.2016.08.021

Gatsé Ebotehouna, C., Xie, Y., Adomako-Ansah, K., Gourcerol, B., Qu, Y., 2021. Depositional Environment and Genesis of the Nabeba Banded Iron Formation (BIF) in the Ivindo Basement Complex, Republic of the Congo: Perspective from Whole-Rock and Magnetite Geochemistry. Minerals 11, 579.

Gourcerol, B., Blein, O., Chevillard, M., Callec, Y., Boudzoumou, F., Djama, L.-M.J., 2022. Depositional Setting of Archean BIFs from Congo: New Insight into Under-Investigated Occurrences. Minerals 12, 114.

Houketchang Bouyo, M., Penaye, J., Mouri, H., Toteu, S.F., 2019. Eclogite facies metabasites from the Paleoproterozoic Nyong Group, SW Cameroon: Mineralogical evidence and implications for a high-pressure metamorphism related to a subduction zone at the NW margin of the Archean Congo craton. J. Afr. Earth Sci. 149, 215–234. https://doi.org/10.1016/j.jafrearsci.2018.08.010

Kwamou Wanang, Kouankap Nono, K.N.G., Nkouathio, D.G., Ayonta Kenne, P., 2021. Petrogenesis and U–Pb zircon dating of amphibolite in the Mewengo iron deposit, Nyong series, Cameroon: fingerprints of iron depositional geotectonic setting. Arab. J. Geosci. 14, 872. https://doi.org/10.1007/s12517-021-07235-8

Lerouge, C., Cocherie, A., Toteu, S.F., Penaye, J., Milési, J.-P., Tchameni, R., Nsifa, E.N., Mark Fanning, C., Deloule, E., 2006. Shrimp U–Pb zircon age evidence for Paleoproterozoic sedimentation and 2.05Ga syntectonic plutonism in the Nyong Group, South-Western Cameroon: consequences for the Eburnean–Transamazonian belt of NE Brazil and Central Africa. J. Afr. Earth Sci., The Precambrian of Central Africa 44, 413–427. https://doi.org/10.1016/j.jafrearsci.2005.11.010

Maurizot, P., Abessolo, A., Feybesse, J.L., Lecomte, P.J., 1986. Etude de prospection minière du Sud-Ouest Cameroun: Synthèse des travaux de 1978 à 1985. Rapp. BRGM. 85, CMR 066.

Moudioh, C., Tamehe, L.S., Ganno, S., NzepangTankwa, M., Brando Soares, M., Ghosh, R., Kankeu, B., Nzenti, J.P., 2020. Tectonic setting of the Bipindi greenstone belt, northwest Congo craton, Cameroon: Implications on BIF deposition. J. Afr. Earth Sci. 171, 103971. https://doi.org/10.1016/j.jafrearsci.2020.103971

Mvodo, H., Ganno, S., Kouankap Nono, G.D., Fossi, D.H., NgaEssomba, P.E., NzepangTankwa, M., Nzenti, J.P., 2022. Petrogenesis, LA-ICP-MS zircon U-Pb geochronology and geodynamic implications of the Kribimetavolcanic rocks, Nyong Group, Congo craton. ActaGeochim. 1–26.

Ndema Mbongue, J.L., Ngnotue, T., Ngo Nlend, C.D., Nzenti, J.P., Cheo Suh, E., 2014. Origin and evolution of the formation of the Cameroon Nyong Series in the western border of the Congo Craton. J. Geosci. Geomat. 2, 62–75.

Ndime, E.N., Ganno, S., Nzenti, J.P., 2019. Geochemistry and Pb–Pb geochronology of the Neoarchean Nkout West metamorphosed banded iron formation, southern Cameroon. Int. J. Earth Sci. 108, 1551–1570.

Nga Essomba Tsoungui, P., Ganno, S., Tanko Njiosseu, E.L., Ndema Mbongue, J.L., Kamguia Woguia, B., SohTamehe, L., Takodjou Wambo, J.D., Nzenti, J.P., 2020. Geochemical constraints on the origin and tectonic setting of the serpentinized peridotites from the Paleoproterozoic Nyong series, Eseka area, SW Cameroon. ActaGeochim. 39, 404–422. https://doi.org/10.1007/s11631-019-00368-4

Nzepang Tankwa, M., Ganno, S., Okunlola, O.A., TankoNjiosseu, E.L., SohTamehe, L., Kamguia Woguia, B., Mbita, A.S.M., Nzenti, J.P., 2021. Petrogenesis and tectonic setting of the Paleoproterozoic KelleBidjoka iron formations, Nyong group greenstone belts, southwestern Cameroon. Constraints from petrology, geochemistry, and LA-ICP-MS zircon U-Pb geochronology. Int. Geol. Rev. 63, 1737–1757.

Owona, S., Schulz, B., Minyem, D., Ratschbacher, L., Tchamabe, B.C., Olinga, J.B., Ondoa, J.M., Ekodeck, G.E., 2022. Eburnean/Trans-Amazonian orogeny in the Nyong complex of southwestern Cameroon: Meta-basite geochemistry and metamorphic petrology. J. Afr. Earth Sci. 190, 104515.

Soh Tamehe, L., Chongtao, W., Ganno, S., Simon, S.J., Kouankap Nono, G.D., Nzenti, J.P., Lemdjou, Y.B., Lin, N.H., 2019. Geology of the Gouap iron deposit, Congo craton, southern Cameroon: Implications for iron ore exploration. Ore Geol. Rev. 107, 1097–1128. https://doi.org/10.1016/j.oregeorev.2019.03.034

Soh Tamehe, L., Li, H., Ganno, S., Chen, Z., Lemdjou, Y.B., Elatikpo, S.Y., 2022. Insight into the Origin of Iron Ore based on Elemental Contents of Magnetite and Whole-Rock Geochemistry: a Case of the Bipindi Banded Iron Formations, Nyong Complex, SW Cameroon. J. Earth Sci., in press. https://kns.cnki.net/kcms/detail/42.1788.P.20220124.1857.014.html

Soh Tamehe, L., Nzepang Tankwa, M., Chongtao, W., Ganno, S., Ngnotue, T., Kouankap Nono, G.D., Simon, S.J., Zhang, J., Nzenti, J.P., 2018. Geology and geochemical constrains on the origin and depositional setting of the Kpwa–Atog Boga banded iron formations (BIFs), northwestern Congo craton, southern Cameroon. Ore Geol. Rev. 95, 620–638. https://doi.org/10.1016/j.oregeorev.2018.03.017

Soh Tamehe, L., Wei, C., Ganno, S., Rosière, C.A., Nzenti, J.P., Gatsé Ebotehouna, C., Lu, G., 2021. Depositional age and tectonic environment of the Gouap banded iron formations from the Nyong group, SW Cameroon: Insights from isotopic, geochemical and geochronological studies of drillcore samples. Geosci. Front. 12, 549–572. https://doi.org/10.1016/j.gsf.2020.07.009

Teutsong, T., Bontognali, T.R.R., Ndjigui, P.-D., Vrijmoed, J.C., Teagle, D., Cooper, M., Vance, D., 2017. Petrography and geochemistry of the Mesoarchean Bikoula banded iron formation in the Ntem complex (Congo craton), Southern Cameroon: Implications for its origin. Ore Geol. Rev. 80, 267–288. https://doi.org/10.1016/j.oregeorev.2016.07.003